computational biology

protein structural classification,
protein structure networks, graphlets

**Author for correspondence:**
Tijana Milenković
e-mail: tmilenko@nd.edu

†These authors contributed equally to this study.

# Network-based protein structural classification

Khalique Newaz[1,2,3], Mahboobeh Ghalehnovi[1,†],
Arash Rahnama[4,†], Panos J. Antsaklis[4]
and Tijana Milenković[1,2,3]

[1]Department of Computer Science and Engineering, [2]Center for Network and Data Science, [3]Eck institute for Global Health, and [4]Department of Electrical Engineering, University of Notre Dame, Notre Dame, IN 46556, USA

 KN, 0000-0002-1192-8360; TM, 0000-0002-8023-6907

Experimental determination of protein function is resource-consuming. As an alternative, computational prediction of protein function has received attention. In this context, protein structural classification (PSC) can help, by allowing for determining structural classes of currently unclassified proteins based on their features, and then relying on the fact that proteins with similar structures have similar functions. Existing PSC approaches rely on sequence-based or direct three-dimensional (3D) structure-based protein features. By contrast, we first model 3D structures of proteins as protein structure networks (PSNs). Then, we use network-based features for PSC. We propose the use of graphlets, state-of-the-art features in many research areas of network science, in the task of PSC. Moreover, because graphlets can deal only with unweighted PSNs, and because accounting for edge weights when constructing PSNs could improve PSC accuracy, we also propose a deep learning framework that automatically learns network features from weighted PSNs. When evaluated on a large set of approximately 9400 CATH and approximately 12 800 SCOP protein domains (spanning 36 PSN sets), the best of our proposed approaches are superior to existing PSC approaches in terms of accuracy, with comparable running times. Our data and code are available at https://doi.org/10.5281/zenodo.3787922

## 1. Introduction

### 1.1. Motivation and related work

Proteins are major molecules of life, and thus understanding their cellular function is important. However, doing so experimentally is costly and time-consuming [1]. Instead, computational approaches

†These authors contributed equally to this study.

are often used for this purpose, which are much more efficient because they leverage on the fact that (sequence or three-dimensional (3D)) structural similarity of proteins often indicates their functional similarity [2]; note that here we refer to as broad notion of protein structural similarity as possible, i.e. as captured by any existing method. One type of such computational approaches is *protein structural classification* (PSC) [3]. PSC uses structural features of proteins with known labels (typically CATH [4] or SCOP [5] structural classes) to learn a classification model in a supervised manner (i.e. by including the labels into the process of training the model). Then, the structural feature of a protein with unknown label can be used as input to the classification model to determine the structural class of the protein. This information can in turn be used to predict function of a protein based on functions of other proteins that belong to the same structural class as the protein of interest. In this paper, we focus on the PSC problem.

Note that there exists a related computational problem which can help with protein function prediction—that of *protein structural comparison* [6]. However, unlike PSC: (i) protein structural comparison uses structural features of proteins with known or unknown labels in unsupervised rather than supervised manner (i.e. it ignores any potential label information), and (ii) it uses the features to compute pairwise similarities between the proteins in hope that highly similar proteins will have the same label (where the labels are used only after the fact), rather than predicting the label of a single protein. In other words, both the goals and working mechanisms of PSC and protein structural comparison are different. Hence, the two approach categories are not comparable to and cannot be fairly evaluated against each other.

Since proteins with high sequence similarity typically also have high 3D structural and functional similarity, traditional PSC approaches have relied only on sequence-based protein features [7,8]. A popular baseline (i.e. naive) sequence feature is the amino acid composition (AAComposition), which measures relative composition of different amino acid types in a protein sequence [6]. Another popular still baseline though somewhat more comprehensive sequence feature is the di-amino acid composition (DAAComposition), which measures relative composition of different ordered di-amino acid combinations in a protein sequence [9]. Some other, even more comprehensive sequence features include the position-specific scoring matrix (PSSM) [10], three-state secondary structure (SS) profile [11], and hidden Markov model (HMM) profile [12], all of which were recently used by a PSC approach called SVMfold [8]. SVMfold integrates the above three sequence features to represent a protein sequence and then uses support vector machine (SVM) as the classification algorithm to perform PSC. Among the most recent sequence-based PSC approaches are MV-fold [13] and DeepFR [14]. MV-fold integrates four sequence features: the same three sequence features that are also used by SVMfold, plus some physico-chemical information in the form of pseudo amino acid composition (PseAAC). DeepFR predicts contacts between residues of a protein sequence and then uses these predicted contacts along with a deep convolutional neural network for PSC. SVMfold, MV-fold and DeepFR methods were compared against each other in the MV-fold study. MV-fold was comparable to SVMfold and it outperformed DeepFR. So, SVMfold is still among the best sequence-based protein features.

Although sequence features have been extensively used for the purpose of PSC, it has been argued that proteins with low sequence similarity can still show high 3D structural and functional similarity [15]. On the other hand, proteins with high sequence similarity can have low 3D structural and functional similarity [16]. Hence, PSC based on 3D-structural as opposed to (or in addition to) sequence features could more correctly identify the structural class of a protein [17].

Typically, 3D-structural approaches extract features directly from the 3D structures of proteins and then use these direct 3D-structural features (i.e. coordinate positions of the atoms in the 3D space) to compare proteins [17,18]. Interestingly, recent 3D-structural PSC approaches have focused on classification based on protein *pairs* [3,19,20]. For example, they consider a pair of protein structures—one with known label (class) and the other one with unknown label, and if the proteins are similar enough in terms of their 3D-structural (and possibly also sequence) features, they assign the known label of the currently classified protein to the currently unclassified protein. As such, these approaches fall somewhere in-between PSC (because both are supervised, but PSC analyses a single protein at a time) and protein structural comparison (because both focus on protein pairs, but protein structural comparison is unsupervised). Therefore, they are not comparable to and cannot be directly evaluated against approaches that solve the PSC problem as defined in our study.

In addition, several *fully unsupervised* approaches have also been proposed that use 3D-structural features to compare protein structures [21,22]. For example, recently, a 3D-structural feature called tuned Gauss integrals (GIT) [23] was used to cluster proteins into structurally similar groups [21].

In contrast to the direct 3D-structural features, protein 3D structures can first be modelled using *protein structure networks* (PSNs), in which nodes are amino acids and edges link amino acids that are

spatially close enough to each other. Then, network-based features can be extracted from the PSNs and used in the task of PSC. A popular concept in this regard is the notion of protein contact maps [24], which are nothing but an alternative representation of PSNs. A contact map is the representation of an $n$ amino acid-long 3D protein structure into an $n \times n$ 2-dimensional matrix $C$. In a contact map $C$, $C_{ij}$ has a value of 1 if the amino acids $i$ and $j$ are within a predefined distance cutoff, i.e. they are in contact, and 0 otherwise. A recent approach that used contact map-based features for PSC is the cutoff scanning matrix (CSM) [25].

Unlike contact maps that are 'simple' network representations, there exists a different category of PSN features that are based on graph-theoretic concepts, i.e. that measure different network properties. One such baseline PSN feature is *Existing-all*, which integrates seven network properties to represent a PSN [6]. Another popular PSN feature that counts different types of network patterns is the concept of graphlets; graphlets are subgraphs or small lego-like building blocks of complex networks [26].

We believe that the graph-theoretic PSN-based PSC is promising. This is because we recently proposed an unsupervised protein structural comparison approach called GRAFENE that relies on graphlets as PSN features of a protein [6]. Given a set of PSNs as input, GRAFENE first extracts different versions of graphlet features from each PSN. Then, it quantifies structural similarity between each pair of the PSNs by comparing their features in an unsupervised manner. GRAFENE outperformed other state-of-the-art 3D-structural protein comparison approaches, including DaliLite [27] and TM-align [28].

In this work, we use the graphlet-based PSN features for the first time in the task of supervised PSC, with a hypothesis that they will improve upon state-of-the-art non-graphlet PSN features and non-PSN features that have traditionally been used in this task. Note that there exists a supervised approach that used graphlets to study proteins [29]. However, it did so in the task of functional classification of amino acids, i.e. nodes in a PSN, rather than in our task of structural classification of proteins, i.e. PSNs. Also, this approach only used the concept of *regular graphlets*, while we also test a newer concept of *ordered graphlets* [30] (see Methods), which outperformed regular graphlets in the GRAFENE study [6].

In general, a PSC approach comprises two key aspects: (i) a method to extract features from a protein structure, and (ii) selection of a classification algorithm to be trained based on the features (and protein labels). Hence, existing PSC approaches can be divided into two broad categories. The first category includes approaches that focus on improving a classification algorithm by relying on existing features [31–33]. The second category includes approaches that extract novel features to predict the structural class of a protein by relying on existing classification algorithms [8,34,35]. Our study belongs to the second category, since our goal is to evaluate graphlet features against other state-of-the-art PSC features in a fair evaluation framework, i.e. under the same (representative) classifier, without necessarily aiming to find the best classifier.

## 1.2. Our contributions

We propose a PSC framework called *NETPCLASS* (**net**work-based **p**rotein structural **class**ification). As one part of our framework, we suggest the use of graphlet- and thus PSN-based protein features in the PSC task under an existing classification algorithm. As another part of our framework, we aim to achieve the following. Existing PSN definition links with unweighted edges those pairs of amino acids whose 3D spatial distance is below some predefined cutoff. Current graphlets can deal with such edge-unweighted networks. We hypothesize that PSC can benefit from including the actual spatial distances as edge weights. However, availability of sophisticated network (e.g. graphlet-based) methods for extracting features from *weighted* networks is limited, and developing such methods is a non-trivial task. So, to evaluate this hypothesis, the best we can currently do is to model a PSN as a simple weighted adjacency matrix and then develop a deep learning-based PSC approach to extract features from such a matrix automatically.

More details about our study are as follows:

(i) We evaluate, in the task of PSC, nine versions of graphlet features that were already used for unsupervised protein structural comparison [6]. In addition, in the same task, we evaluate a non-graphlet network (Existing-all) feature [6], a recent contact map-based (CSM) feature [25], a recent 3D-structural (GIT) feature [21], a state-of-the-art sequence (SVMfold) feature, and two baseline sequence features (AAComposition and DAAComposition) [6,9]. We use the same, reasonably chosen, classification algorithm for each of the above 15 features to learn (train) their classification models, in order to fairly compare their performance. By reasonably chosen, we mean a well-performing conventional classification algorithm. It has been observed that in situations

where we already have a well-performing feature under a conventional classification algorithm, non-conventional (e.g. deep learning-based) classification algorithm might not provide that much of a performance boost compared to the conventional one [36]. Also, it is well known that using non-conventional classification algorithm (e.g. deep convolutional neural networks or recurrent neural networks) would come at a cost of much larger computational time. So, we believe that in order to evaluate the meaningfulness of graphlets over the existing state-of-the-art PSC features, which is the key task of our paper, there is no need to use more complex classification algorithms. Here, as a proof-of-concept, we use a simple yet powerful *logistic regression* (LR) classifier, whose output indicates, for the given input protein and each class, the likelihood that the protein belongs to the given class. Note that in initial stages of our evaluation, in addition to LR, we considered an SVM classifier. We found that using SVM showed no improvement compared to using LR (under same features and on same PSN data) in terms of accuracy, while at the same time it was slower. So, we had no reason to continue using SVM.

(ii) Since the different categories (i.e. sequence, 3D-structural or PSN-based) of protein structural features can provide complementary information, we combine the individual features to form new integrated features and evaluate these against each of the individual features.

(iii) Because graphlets are currently designed only for unweighted PSNs, and because the current literature lacks knowledge on how to efficiently extract meaningful features from a weighted network, we aim to extract such features automatically via *deep learning* (DL).

(iv) We evaluate the considered approaches on a large set of approximately 9400 CATH and approximately 12 800 SCOP protein domains. We transform protein domains to PSNs with labels corresponding to CATH and SCOP structural classes, where we study each of the four levels of CATH and SCOP hierarchies [6]. Our evaluation is based on measuring how correctly the trained classification models can predict the classes of labelled proteins in the test data using 10-fold cross-validation.

Our key findings are as follows.

In terms of PSC accuracy, when we compare the individual features, we observe that the best of our graphlet features outperform all of the other individual features except GIT and SVMfold. However, regarding GIT, while it shows somewhat superior performance to the graphlet features, GIT is only applicable to proteins, while graphlets are general-purpose network features and are applicable to many other complex systems that can be modelled as networks. Further, we show that integrating GIT with the best graphlet feature significantly improves accuracy of each of GIT alone and the graphlet feature alone, which means that each of the two individual features contributes to the superior performance of their integrated version. Note that we find that integrating the best of the sequence features with GIT and graphlets does not yield significant improvements. Regarding SVMfold, while this approach performs well (comparable to the best of our proposed approaches), SVMfold is orders of magnitude slower than our proposed approaches. In fact, SVMfold is so slow that we were able to run it only on 5.5% of our data. In terms of running time, SVMfold is the slowest, followed by the integrated features, followed by CSM, and followed by the rest of the features (which are mostly comparable to each other).

Accounting for edge weights in PSNs via DL achieves accuracy that is relatively comparable to accuracy of the individual unweighted network-based methods. Note that here we are comparing as simple as possible weighted network features (the simple weighted adjacency matrix) against highly sophisticated unweighted network features (graphlets, which are the state-of-the-art in network science). So, a comparable accuracy of the former and the latter implies a promise of future weighted network-based analyses of protein 3D structures (such as developing and using weighted graphlet-based features).

# 2. Methods

## 2.1. Data and protein structure network construction

*First*, we use a set of 17 036 proteins that was previously used in the large-scale unsupervised protein structural comparison GRAFENE study [6]. Per this study, this dataset contains all proteins from the Protein Data Bank (PDB) [37] that are at most 90% sequence similar. To identify protein domains, we use two protein domain categorization databases: CATH and SCOP. Note that we can only use those CATH and SCOP protein domains that are present in our set of 17 036 proteins.

To construct a PSN from a protein domain, we use the PDB files [37], which contain information about the 3D coordinates of the heavy atoms (i.e. *carbon*, *nitrogen*, *oxygen* and *sulfur*) of the amino acids in the domain. In a PSN, nodes are amino acids of a protein domain and there is an edge between any two nodes if they are sufficiently close in the 3D space. Clearly, given a protein domain, its corresponding PSN construction depends on (i) the choice of atom(s) of an amino acid to represent it as a node in the PSN, and (ii) a distance cutoff between a pair of nodes to capture their spatial proximity. There is no established knowledge about what choices of node representation and distance cutoff are the best to construct a PSN. Results of past studies that exclusively evaluated the effect of different choices of node representation and distance cutoff on the meaningfulness of PSNs/contact maps have largely been contradicting [38–41]. For example, in terms of the node representation choice, Duarte *et al.* [39] identified that $\beta$ carbon as a node in a PSN captures the 3D structure of a protein more accurately than the use of $\alpha$ carbon, implying that the choice of which atom type to consider as a node in a PSN matters. On the other hand, da Silveira *et al.* [38] found that the choice of which atom type to consider as a node in a PSN does not make a difference. Similarly, in terms of the choice of distance cutoff, while Vassura *et al.* [42] showed that an increase in distance cutoff generally increased the meaningfulness of PSNs, Duarte *et al.* [39] showed that this is not necessarily the case. So, there seems to exist no consensus on exactly how to best choose a node representation or distance cutoff value for PSN construction. Nonetheless, our choice of these parameter values is made somewhat easier in the light of a recent study on unsupervised protein structural comparison that used the exact same data as we do here [6]. By considering four different combinations of atom choice and distance cutoff definitions (any heavy atom with 4, 5 and 6 Å distance cutoffs and $\alpha$-carbon with 7.5 Å distance cutoff), this study showed that the choice of atom and distance cutoff did not significantly affect the overall protein structural comparison performance [6]. Motivated by this, for most of our analyses, i.e. unless explicitly indicated otherwise, we consider only one of these PSN construction strategies in our study: we define an edge between two amino acids if the spatial distance between any of their heavy atoms is within 4 Å. Because the choice of a PSN construction strategy only affects PSN-based features, and because as we will show later that the best of our graphlet features is the best among the considered PSN-based features, for it, we also use the distance cutoff of 6 Å; we will explicitly note whenever 6 Å is used instead of 4 Å. Note that we also considered the cutoff of 5 Å, but using 6 Å was always superior to using 5 Å. Hence, for simplicity, we do not report results for 5 Å.

Additionally, for most of our analyses, i.e. unless explicitly indicated otherwise, in order to only keep 'meaningful' PSNs for further analysis, we filter the PSNs using an established guideline that is based on network properties of a PSN [6]. Namely, we only keep a PSN if it has (i) a single connected component, (ii) a diameter of at least six, and (iii) at least 100 nodes (amino acids) (electronic supplementary material, section S1). Following these criteria, we obtain 9509 and 11 451 PSNs corresponding to CATH and SCOP, respectively. Note that in order to fairly compare the considered protein features to each other, we want to focus on only those protein domains (i.e. PSNs) that can be processed by *each of the features*. Removal from the above data of those protein domains that cannot be processed by at least one considered feature results in 9440 CATH and 11 352 SCOP PSNs for further analysis. These are the final data that are used throughout the study, unless explicitly told otherwise. Importantly, in order to evaluate the effect of the above three PSN filtering criteria on our results, for a subset of our analyses, we consider even those connected PSNs that have diameter less than six or fewer than 100 nodes; we use the resulting data only in §3.8.

Given the CATH PSN data, we do the following. First, we test the power of the considered PSC approaches to predict the following top hierarchical level classes of CATH: *alpha* ($\alpha$), *beta* ($\beta$) and *alpha/beta* ($\alpha/\beta$). Note that *few secondary structures* is the remaining CATH top hierarchical class, but none of the CATH PSNs belongs to this class, so we do not consider it. Hence, we take all 9440 CATH PSNs and identify them as a single PSN set, where the PSNs have labels corresponding to three top level CATH classes: $\alpha$, $\beta$, and $\alpha/\beta$. Second, we compare the approaches on their ability to predict the second level classes of CATH, i.e. within each of the top-level classes, we classify PSNs into their sub-classes. To ensure enough training data, we focus only on those top-level classes that have at least two sub-classes with at least 30 PSNs each; we require the minimum of 30 PSNs in order to have sufficient statistical power in the classification task [43]. Three classes satisfy this criteria. For each such class, we take all of the PSNs belonging to that class and form a PSN set, which results in three PSN sets. Third, we compare the approaches on their ability to predict the third-level classes of CATH, i.e. within each of the second-level classes, we classify PSNs into their sub-classes. Again, we focus only on those second-level classes that have at least two sub-classes with at least 30 PSNs each. Nine classes satisfy this criteria. For each such class, we take all of the PSNs belonging to that class and form a PSN set, which results in nine PSN

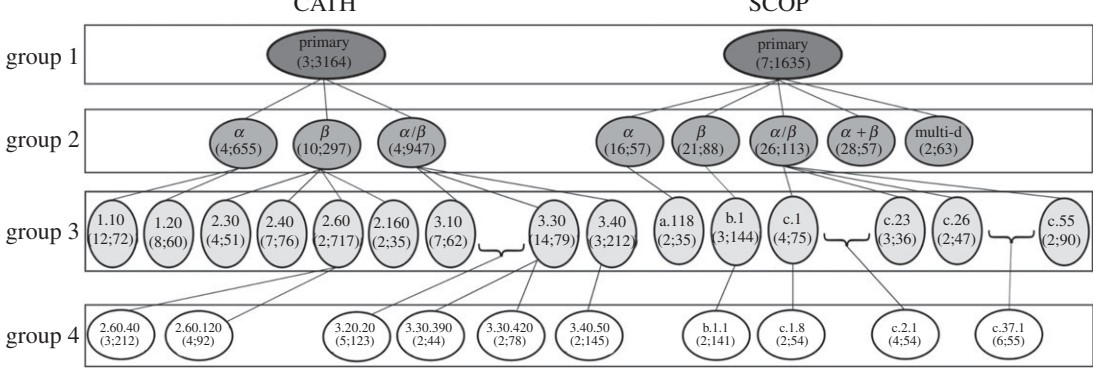

**Figure 1.** Hierarchical representation of the 35 CATH and SCOP PSN sets that we use. Each oval shape represents a PSN set. The top line in the given oval indicates the name of the PSN set. The bottom line in the given oval contains two numbers represented as '$x$; $y$', where $x$ is the number of classes (i.e. labels) that are present in the PSN set and $y$ is the average number of PSNs per class. For example, for the SCOP database, PSN set SCOP-primary has seven classes, where each class has an average of 1635 PSNs. All of the PSN sets at a given level form a PSN set group. For example, PSN sets CATH-primary and SCOP-primary form PSN set group 1. A class of a PSN set in group $i$ may be present as a PSN set in group $i + 1$. For example, the class $\alpha$ of the PSN set SCOP-primary in group 1, is present as a PSN set SCOP-$\alpha$ in group 2. Note that since we select a PSN set if and only if it has at least two classes each with at least 30 PSNs, given a PSN set in group $i$, not all of its classes are necessarily present as PSN sets in group $i + 1$. For example, PSN set SCOP-primary has seven classes in group 1, but only five of its classes exist as PSN sets in group 2. Also note that because of our PSN set selection criteria, it is not necessary that a PSN set in group $i + 1$ has to be present as a class within a PSN set in group $i$. For example, PSN set SCOP-c.2.1, which is present in group 4, is not present as a class within any PSN set in group 3. This is because SCOP-c.2 contains only one class that has at least 30 PSNs (i.e. c.2.1) and hence we do not consider SCOP-c.2 as a PSN set in our analysis. This figure has been adapted from the GRAFENE paper [6]. Note that the numbers of PSNs in this figure are different from the numbers of PSNs in the corresponding figure of the GRAFENE paper, because in this study we focus only on those PSNs that can be analysed by each of the considered protein features (§2.1), where not all of our considered features are necessarily the same as all of the methods considered in the GRAFENE paper.

sets. Fourth, we compare the approaches on their ability to predict the fourth-level classes of CATH, i.e. within each of the third-level classes, we classify PSNs into their sub-classes. We again focus only on those third-level classes that have at least two sub-classes with at least 30 PSNs each. Six classes satisfy this criteria. For each such class, we take all of the PSNs belonging to that class and form a PSN set, which results in six PSN sets.

*Thus, in total, we analyse 1 + 3 + 9 + 6 = 19 CATH PSN sets.* For further details on the number of PSNs and the number of different protein structural classes in each of the PSN sets, see electronic supplementary material, tables S1–S3. We follow the same procedure for the SCOP PSN data *and obtain 1 + 5 + 6 + 4 = 16 SCOP PSN sets.* For more details, see electronic supplementary material, section S2 and tables S1–S3.

We group each of the 19 CATH PSN sets and 16 SCOP PSN sets into four PSN set groups, corresponding to the four hierarchy levels of CATH and SCOP, respectively: group 1 (all 1 + 1 = 2 PSN sets), group 2 (all 3 + 5 = 8 PSN sets), group 3 (all 9 + 6 = 15 PSN sets) and group 4 (all 6 + 4 = 10 PSN sets) (figure 1).

*Second*, in addition to the 19 + 16 = 35 CATH and SCOP PSN sets from the GRAFENE study as described above, we use a different dataset because of the following reason. Typically, high sequence similarity of proteins indicates their high structural similarity. Hence, given a set of proteins in which proteins in the same structural class have high sequence similarity (typically more than 40%), a 'simple' protein sequence comparison might be sufficient to perform PSC [44]. So, we aim to evaluate how well our considered protein features (that are based on different aspects of a protein structure) can identify proteins in the same structural category when all of the proteins (within and across structural categories) show low (less than or equal to 40%) sequence similarity.

In order to do this, we download the dataset called Astral from the SCOPe 2.04 database [45]. This dataset has 14 666 protein domains, where each domain pair is at most 40% sequence similar to each other. Each protein domain in this dataset is annotated by a label (i.e. protein structural class) assigned by the protein domain categorization database SCOP, where the label indicates the protein family to which the domain belongs. We create a PSN corresponding to each of the protein domains

as described above. Then, we follow the same criteria as in the GRAFENE study [6] (also described above) to only keep 'meaningful' PSNs. This results in 1677 PSNs belonging to 32 different protein structural classes. We name this set of 1677 PSNs as *Astral*. Note that the decrease from 14 666 to 1677 protein domains is mostly due to removing protein domains that belong to a protein domain class with fewer than 30 protein domains (see above); approximately 60% of the data loss is caused by this.

Taken together, in our study, we use $35 + 1 = 36$ PSN sets (35 CATH and SCOP PSN sets and one Astral PSN set) that contain 9440 protein domains annotated by the CATH database and 12 820 protein domains (union of the above SCOP-related 11 352 protein domains and the Astral-related 1677 protein domains) annotated by the SCOP database.

## 2.2. Our evaluation framework

### 2.2.1. Protein features

For each of the protein domains, we extract 18 different protein features that are based on either sequence, 3D structure, contact map, non-graphlet network, graphlet network or weighted network (table 1). To understand what aspects of the protein structure each considered feature captures, we categorize the given feature based on whether it (i) captures local or global structure of a protein, (ii) is based on the backbone or the side-chain of a protein, (iii) is dependent on the sequentiality of amino acids of a protein or not, and (iv) is dependent on the 3D-structural conformation of a protein or not. We say that a feature is local if it *explicitly* uses local structural characteristics of a protein to summarize the whole protein structure, while we say that a feature is global if it summarizes the structure of a protein as a whole without explicitly using smaller structural characteristics of a protein. We say that a feature is based on the backbone (side-chain) of a protein if it uses only the heavy atoms of the protein backbone (side-chain). We say that a feature is sequence-dependent if permuting sequence positions of the amino acids of a given protein alters the feature as well. We say that a feature is 3D structural conformation-dependent if altering 3D-structural positions of amino acids of a given protein alters the feature as well. By altering the 3D-structural position of an amino acid, we mean changing the 3D spatial position of the amino acid to any other position in the 3D space, which can be different from positions of all other amino acids of the protein.

Note that all of the considered sequence-based features are not dependent on the 3D-structural conformation of a protein, while all of the other considered features are dependent on this. Because we just covered this, we do not comment any further on whether a feature is 3D conformation-dependent or not.

Also, note that since we construct a PSN using any heavy atoms irrespective of whether the heavy atoms belong to the backbone or the side-chain of a protein (see above), any considered feature that is entirely or partially PSN-based (which is the case for each considered feature except sequence-based AAComposition, DAAComposition and SVMfold, plus 3D structure-based GIT) is automatically based on both the backbone and the side-chain of a protein. Henceforth, we only comment on whether a feature is either backbone-based or side-chain-based (or neither) only if the feature is not PSN-based, i.e. we do it only for AAComposition, DAAComposition, SVMfold and GIT.

Below, we define each of the features and outline whether the given feature is local or global, as well as whether it is dependent on the sequentiality of amino acids of a protein (and for AAComposition, DAAComposition, SVMfold and GIT, whether the given feature is either backbone-based or side-chain-based (or neither)).

**Sequence-based features.** We use a baseline sequence feature, *AAComposition*. Given a protein sequence, AAComposition measures the relative frequency of the 20 types of amino acids in the entire sequence: for each amino acid type $i$, it measures the frequency occurrence of $i$ in the sequence divided by the total number of amino acids in the sequence. AAComposition considers the entire protein sequence, and hence, it is a global feature. AAComposition does not use any heavy atom of an amino acid, and hence, it is neither a backbone- nor a side-chain-based feature. AAComposition only measures the relative frequency of the different amino acid types without looking at the sequence position of a given amino acid, and hence, it is not sequence-dependent.

We use another baseline sequence feature, DAAComposition. Given a protein sequence, DAAComposition measures the relative frequency of each of the possible ordered di-amino acid combinations in the entire sequence: for each pair of amino acid types $i$ and $j$, it measures the frequency occurrence of a given ordered combination of $i$ and $j$ in the sequence divided by the total

**Table 1.** Summary of all protein features that we use in this study. We use the same classifier (logistic regression) for all features except 'distance matrix' (coloured in grey); for the latter, we use deep learning.

| category | feature name | measures local/global structure | based on protein backbone/side-chain | dependent on sequentiality of amino acids | dependent on 3D conformation |
|---|---|---|---|---|---|
| sequence | AAComposition | global | none | no | no |
| | DAAComposition | global | none | yes | no |
| | SVMfold | local | none | yes | no |
| 3D structure | GIT | local | backbone | yes | yes |
| contact map | CSM | global | both | no | yes |
| non-graphlet network | Existing-all | global | both | no | yes |
| graphlet network | Graphlet-3-4 | local | both | no | yes |
| | Graphlet-3-5 | local | both | no | yes |
| | NormGraphlet-3-4 | local | both | no | yes |
| | NormGraphlet-3-5 | local | both | no | yes |
| | OrderedGraphlet-3 | local | both | yes | yes |
| | OrderedGraphlet-3-4 | local | both | yes | yes |
| | OrderedGraphlet-3-4(6 Å) | local | both | yes | yes |
| | NormOrderedGraphlet-3 | local | both | yes | yes |
| | NormOrderedGraphlet-3-4 | local | both | yes | yes |
| integrated | GIT+OrderedGraphlet-3-4(6 Å) | local | both | yes | yes |
| | Concatenate | both | both | yes | yes |
| weighted network | distance matrix | global | both | yes | yes |

number of all possible di-amino acid ordered combinations among all 20 amino acid types. DAAComposition considers the entire protein sequence, and hence, it is a global feature. DAAComposition does not use any heavy atom of an amino acid, and hence, it is neither a backbone- nor a side-chain-based feature. Any change in sequence can alter di-amino acid compositions of a protein, and hence, DAAComposition is sequence-dependent.

Note that we evaluated DAAComposition against two additional related features, tri-amino acid composition and PseAAC [46]. We found DAAComposition to consistently and significantly outperform the latter two. Consequently, for simplicity, we do not report any results for tri-amino acid composition and PseAAC.

Additionally, we use a recent state-of-the-art sequence method called SVMfold. Given a protein sequence, SVMfold computes the PSSM [10], three-state SS profile [11] and HMM profile [12] of the protein sequence, and integrates these three features to obtain a single feature representation of the protein [8]. SVMfold relies on HMM profile, which extracts features from subsequences of a protein, and hence, SVMfold is a local feature. SVMfold does not explicitly use any heavy atom of an amino acid, and hence, it is neither a backbone- nor a side-chain-based feature. SVMfold relies on PSSM profile, which extracts features from a sequence alignment of proteins, and hence, SVMfold is sequence-dependent.

Note that there exists another method called SVM-fold [47] (note that the difference between the names of this method and the above SVMfold method is the presence and the absence of '-', respectively). However,

we do not use this method because the focus of its publication was to propose not a novel protein feature but an improved classification algorithm that, given any sequence-based protein feature, can perform PSC. Specifically, the method feeds an existing sequence-based protein feature into string kernels [48], which it then uses to classify proteins. However, as pointed out earlier in §1, the focus of our study is to evaluate different protein features in the task of PSC and not on improving the underlying PSC algorithm. Additionally, this method (i.e. SVM-fold) is more than a decade old approach, while the sequence-based method that we evaluate (i.e. SVMfold) in our paper is a recent approach.

**3D-structural feature.** We use a state-of-the-art 3D-structural feature, *GIT*. Given a protein structure, GIT measures how often, according to the amino acid sequence, the $\alpha$-carbon trace of the protein forms different kinds of local patterns in the 3D space. To measure the number of different patterns, GIT computes 31 different Gauss integrals and uses them as the feature representation of a protein [23]. GIT measures local patterns formed by the $\alpha$-carbon trace of a protein, and hence, it is a local feature. GIT is based on only the $\alpha$-carbons of a protein, which are part of the backbone of a protein, and hence, GIT is a backbone-based feature. GIT is based on the $\alpha$-carbon trace of a protein, which might be affected by a change in sequence positions of amino acids, and hence, GIT is sequence-dependent.

**Contact map-based feature.** We use a recent contact map-based feature, *CSM*. Given a protein structure, CSM first computes 151 contact maps, which are based on 151 distance cutoffs (ranging from 0.0 to 30.0 Å in increments of 0.2 Å). For a given distance cutoff, two amino acids are considered to be in contact if any of their heavy atoms are within the distance cutoff; then, CSM counts the number of amino acid pairs that are in contact at that cutoff. Finally, CSM uses all 151 counts (for the 151 cutoffs) as the 151-dimensional protein feature vector [25]. CSM counts the total number of contacts present in the whole protein, and hence, it is a global feature. CSM counts the number of contacts, which is not affected with a change in sequence positions of amino acids, and hence, CSM is not sequence-dependent.

**Non-graphlet network-based feature.** Here, we use a feature that was shown to outperform many other non-graphlet network-based features in an unsupervised protein comparison task [6]. We denote this feature as *Existing-all*. Given a PSN, Existing-all calculates and integrates seven global network features: average degree, average distance, maximum distance, average closeness centrality, average clustering coefficient, intra-hub connectivity and assortativity. Existing-all uses global network measures to quantify the structure of a protein (i.e. PSN), and hence, it is a global feature. Existing-all extracts features from a PSN, which is not affected with a change in sequence positions of amino acids, and hence, Existing-all is not sequence-dependent.

**Graphlet network-based features.** Graphlets, as originally defined, are small connected induced non-isomorphic undirected $k$-node subgraphs of a network [26]. For example, an edge is the only 2-node graphlet. There exist two 3-node graphlets: a 3-node path and a triangle. Examples of 4-node graphlets are a 4-node path, a square, a square with exactly one diagonal, or a 4-node clique (complete graph). In total, there exist six 4-node graphlets. Typically, due to high computational complexity (i.e. running time) of counting graphlets in large networks, up to 5-node graphlets are studied. There exist 21 5-node graphlets. Consequently, there exist $1 + 2 + 6 + 21 = 30$ 2–5-node graphlets. Given a network, one can count occurrences of the different graphlet types in the network and use these counts as a network feature [26]. Or, one can count how many times a node/edge in the network participates in the different graphlet types and use these counts as a node/edge feature [49,50]. Because in our study we extract graphlet-based features of an entire network rather than of individual nodes/edges, the latter is out of the scope of this study.

In addition to being undirected, graphlets as originally defined are unordered, homogeneous, static and unweighted. More recently, graphlets were extended to their ordered [6], heterogeneous [51] and dynamic [52] counterparts. For explanations of all of these different graphlet types, see Newaz & Milenković [53]. Note that weighted graphlets are still lacking, despite an existing study using a similar terminology for a concept that even the authors of that study acknowledge is entirely unrelated to graphlets as defined in our paper and in general in the field of network science [54].

In this study, we deal with undirected and unweighted networks and thus rely on undirected and unweighted graphlets. However, in one case, we treat our networks as unordered, and in the other case we treat them as ordered. So, we use unordered graphlets (defined above) for the former and ordered graphlets (defined below) for the latter. Specifically, we use graphlets to extract the following network features from our PSNs.

*Graphlet counts.* We use two graphlet-based protein features, i.e. *Graphlet-3-4* and *Graphlet-3-5*. Given a PSN, Graphlet-3-4 and Graphlet-3-5 count the number of 3–4-node and 3–5-node graphlets, respectively.

In particular, in the Graphlet-3-4 or Graphlet-3-5 feature vector, position $i$ represents the logarithm of the count of graphlets of type $i$ [6]. Both Graphlet-3-4 and Graphlet-3-5 are based on graphlets, which are small (i.e. local) network patterns as defined above, and hence, both Graphlet-3-4 and Graphlet-3-5 are local features. Both Graphlet-3-4 and Graphlet-3-5 extract features from a PSN, which is not affected with a change in sequence positions of amino acids, and thus, neither Graphlet-3-4 or Graphlet-3-5 are sequence-dependent.

*Normalized graphlet counts.* Since PSNs can be of very different sizes, we use two recent protein features that are based on normalized graphlet counts and that thus account for network size differences [6]. These features are *NormGraphlet-3-4* and *NormGraphlet-3-5*; they are normalized equivalents of Graphlet-3-4 and Graphlet3-5, respectively. In particular, given a PSN, in both NormGraphlet-3-4 and NormGraphlet-3-5 feature vectors, a position $i$ represents the total count of graphlets of type $i$ divided by the sum of the counts of all graphlet types. Similar to Graphlet-3-4 and Graphlet-3-5, NormGraphlet-3-4 and NormGraphlet-3-5 are local and sequence-dependent features.

*Ordered graphlet counts.* Graphlets capture 3D-structural but not sequence information. To integrate the two, ordered graphlets were proposed [30]. These are graphlets whose nodes acquire a relative ordering based on positions of the amino acids in the sequence. Two ordered graphlet features exist: *OrderedGraphlet-3* and *OrderedGraphlet-3-4* [6,30]. For a PSN, in OrderedGraphlet-3 and OrderedGraphlet-3-4 feature vectors, position $i$ is the total count of ordered graphlets of type $i$.

In addition, we use two features that are based on normalized counts of ordered graphlets [6]: *NormOrderedGraphlet-3* and *NormOrderedGraphlet-3-4*; these are normalized equivalents of OrderedGraphlet-3 and OrderedGraphlet-3-4, respectively. For a PSN, in NormOrderedGraphlet-3 and NormOrderedGraphlet-3-4 feature vectors, position $i$ is the total count of ordered graphlets of type $i$ divided by the total count of all ordered graphlet types.

As we will show, OrderedGraphlet-3-4, when used at the default 4 Å distance cutoff (§2.1), is the best of all considered (graphlet or non-graphlet) PSN-based features. So, for this best PSN feature, we aim to evaluate the effect of the choice of distance cutoff. Consequently, we also count ordered graphlets in PSNs that are based on the 6 Å distance cutoff, resulting in a new feature called *OrderedGraphlet-3-4(6 Å)*.

Just like regular graphlets, ordered graphlets are local network structures, and hence, all of the ordered graphlet-based features are local. The order of nodes in ordered graphlets is determined by sequence positions of amino acids in a protein, and hence, all of the ordered graphlet-based features are sequence-dependent.

**Integrated features.** We propose two new features that integrate two different subsets of the above protein features. First, we integrate the best of the non-graphlet features (GIT) and the best of the graphlet features (OrderedGraphlet-3-4(6 Å)) (§3) into a new combined feature called *GIT+OrderedGraphlet-3-4(6 Å)*. Second, we add the best of the sequence features that we could run on all of the data (i.e. DAAComposition) to GIT+ OrderedGraphlet-3-4 (6 Å) into a new combined feature called *Concatenate*. That is, even though SVMfold is better than DAAComposition on the data where we were able to run SVMfold, we use DAAComposition instead of SVMfold because of SVMfold's high running time and thus its inability to be run on all data (§3). Formally, given $k$ features of size $1 \times f_i$ ($i = 1, 2, \ldots, k$), we concatenate them to form a feature of size $1 \times \sum_{i=1}^{k} f_i$. Clearly, any integrated feature will belong to all of those categories to which the corresponding individual features belong (table 1).

**Principal component analysis (PCA)-transformed features.** For each of the 17 features described above (nine graphlet features, one non-graphlet network feature, one contact map feature, one 3D-structural feature, three sequence features, and two integrated features), we aim to test whether it is possible to improve effectiveness of the features by reducing dimensions of their vectors in order to remove potential redundancies between the different vector dimensions. We perform feature dimensionality reduction using PCA for the following reasons. PCA transformation of protein features, including graphlet-based ones, was shown to better capture protein (dis)similarity in the unsupervised task of protein structural comparison [6]. Also, PCA, a linear feature dimensionality reduction method, was shown to work better than two well-known nonlinear methods, minimum curvilinear embedding (MCE) and non-centred MCE, when graphlet features were used to compare (align) biological networks [51].

We perform the PCA feature transformation for each of the 17 features and each of the 36 PSN sets as follows. Let us denote a given feature's vector dimension as $N$. Let us denote a given PSN set's number of proteins as $M$. The $N$-dimensional feature vectors over all $M$ proteins in the set result in an $M \times N$ matrix. We apply PCA to this matrix. That is, we reduce the original feature vectors of length $N$ to new feature vectors of length $r$ ($r \leq N$), where the latter correspond to the first $r$ principal components (i.e. the first $r$ eigenvalues). We set the value of $r$ to be at least two and as low as possible so that the $r$

components account for at least 90% of variation in the PSN set. We use *prcomp()* function in R to PCA-transform the original $M \times N$ matrix.

**Weighted network-based feature.** We use a weighted adjacency matrix, or distance matrix [55], of a 3D protein structure as a weighted PSN-based feature representation. In particular, given a protein of length $n$, we define a weighted adjacency matrix $D$ of size $n \times n$, in which each position $D_{ij}$ contains the minimum 3D spatial distance between the amino acids $i$ and $j$, where the minimum is taken over all pairwise distances between any heavy atoms of $i$ and $j$. Weighted adjacency matrix measures pairwise distances among amino acids in the entire protein, and hence, it is a global feature. A change in sequence positions of amino acids will alter the matrix values, and hence, weighted adjacency matrix is sequence-dependent.

Taken together, in our study, we use 35 different protein features (15 individual pre-PCA features, 15 individual post-PCA features, two integrated pre-PCA features, two integrated post-PCA features and a weighted network-based feature).

## 2.2.2. The logistic regression framework

Given a PSN set, we train a logistic regression (LR) classifier corresponding to each of 34 out of the 35 pre- or post-PCA protein features (see above); we use a different classifier for the remaining (weighted network-based) feature, as discussed in the following subsection. Hence, for each of the PSN sets, we get 34 different trained LR classifiers. In each of the trained classifiers, the input is a feature representation of a protein and output is the structural class to which the protein belongs. Since PSC is a multi-class problem, we use the one-vs-rest scheme to train an LR classifier. Due to space constraints, we provide further details about our LR framework in electronic supplementary material, section S3.

Given a PSN set and a protein feature, we first divide the PSN set into 10 equal-sized subsets, such that each subset contains the same proportion of different protein structural classes (i.e. labels) as present in the initial PSN set. Then, for each subset, we use it as the test data and the union of the remaining nine subsets as the training data. We use the training data in two different ways to train an LR model. First, we train an LR model with the training data as is. We call this way of training an LR model as *proportional*. Second, at least some of our PSN sets have unequal numbers of proteins in different classes, i.e. are imbalanced. Consequently, the training data as used by the proportional approach is also unbalanced. Hence, we exploit a data re-sampling approach called synthetic minority oversampling technique (SMOTE) [56] to first balance the training data and only then use it to train an LR model. We call this way of training an LR model as *proportional+SMOTE*.

Before we train an LR model and use it to predict classes of proteins in the test data, we use the training data itself in a 10-fold cross validation manner [57,58] to choose an 'optimal' value for the regularization hyper-parameter of an LR model. We perform linear search to find an optimal value from the set $[2^{-2}, 2^{-1}, 2^0, 2^1, 2^2]$. For details, see electronic supplementary material, section S3. Then, we use the resulting optimal hyper-parameter value and all of the training data to train an LR model.

Next, we evaluate the performance of the trained model on the separate test data, and we do so using two evaluation measures: accuracy and Matthew's correlation coefficient (MCC) [59]. Accuracy is the percentage of all proteins from the test data that are classified into their correct protein structural classes. In case of an imbalanced dataset, accuracy can give over-optimistic classifier performance, as its value is biased towards the class with the largest number of samples. That is, accuracy can fail to capture how a classifier performs on each of the classes present in the dataset. So, we also use MCC, which intuitively captures how well a given classifier performs on each of the classes and hence is robust to data imbalance. Because we measure the performance of a trained LR model over 10 different sets of test data, we report an accuracy or MCC performance value averaged over the 10 sets.

## 2.2.3. The deep learning framework

In the second part of our study, we design a deep learning (DL) framework in order to learn features of 3D protein structures using weighted protein structure networks. For each of the 36 PSN datasets, we train a deep neural network, where we use distance matrix-representations of proteins as input. Our DL framework consists of one input layer, seven hidden layers, and an output layer. Due to space constraints, we provide further details about our DL framework in electronic supplementary material, section S4.

Similar to LR framework, given a PSN set and a protein feature, we first divide the PSN set into 10 equal-sized subsets, such that each subset contains the same proportion of different protein

structural classes (i.e. labels) as present in the initial PSN set. Then, for each subset, we use it as the test data and the union of the remaining nine subsets as the training data to train our DL framework. Here, we train using only the proportional approach but not proportional+SMOTE, because (as we will show) our evaluation in the LR framework reveals that results are qualitatively identical no matter whether proportional or proportional+SMOTE training is used.

Note that recently a related deep learning method was proposed that uses 3D-structural information for protein function prediction [60]. Given a protein, the method extracts two types of structural information: the torsional angles for each of the amino acids and the pairwise spatial distances between the $\alpha$-carbons of the amino acids of a protein. Given this structural information, the method uses a deep convolutional neural network framework along with SVM to perform the protein function prediction. The method was applied to classify enzymes (i.e. proteins) into functional categories. Since we only became aware of this very recent method towards the end of our work, we could not include it into our evaluation. Also, this method extracts 3D-structural features rather than PSN-based features, but we already compare to a state-of-the-art 3D-structural feature, GIT, and so including an additional method of the same type is not critical. Additionally, the focus of the study that proposed this existing 3D-structural method was on the task of protein function prediction rather than on our task of PSC, while GIT, the 3D-structural feature that we already consider, was proposed for protein structural (rather than functional) analyses and is thus much more related to our proposed work.

# 3. Results and discussion

Throughout this section, unless stated otherwise, we: (i) analyse the 35 considered PSN sets that span all four levels (groups) of CATH and SCOP hierarchies, plus the Astral PSN set, totalling to 36 PSN sets; (ii) report results when using the proportional approach in the training stage of the classification process and the accuracy measure; and (iii) for the PSN-based methods, which depend on the choice of distance cutoff, we use the default 4 Å cutoff.

In §3.1, we compare the different graphlet features (§2.2) to identify the best one(s) for further analyses. In §3.2, we identify the best of the pre- or the post-PCA versions for each of the considered features. In §3.3, we evaluate how well the best of the graphlet feature(s) perform in comparison to the existing baseline or state-of-the-art protein features that we study (considering for each feature the best of its pre- and post-PCA versions). Here, we leave out from consideration the existing SVMfold sequence approach [8], because we were unable to apply this approach to all 36 considered PSN sets due to its extremely high running time. Instead, we consider the SVMfold later on, in a smaller-scope analysis of two of the 36 PSN sets on which we were able to run SVMfold (see below). In §3.4, we evaluate whether integration of different feature types improves upon each of the individual features. In §3.5, we analyse whether results from the above analyses differ when using the proportional vs proportional+SMOTE approach in the training stage of the classification process as well as when using the accuracy vs MCC measure. In §3.6, we compare the performance of the best graphlet-based PSC approaches that deal with unweighted PSNs to the performance of simple weighted PSN-based feature classification via DL. In §3.7, we analyse two representative PSN sets on which SVMfold could be run, in order to compare our proposed approaches to this state-of-the-art existing sequence-based PSC approach. In §3.8, we study the effect of the methodological choice of considering versus not considering our PSN filtering criteria (from §2.1) on the performance of the considered features for one of the considered PSN sets.

## 3.1. Comparison of graphlet features

When we compare all graphlet features (that use the default 4 Å distance cutoff) under the LR classifier, OrderedGraphlet-3-4 is consistently and by far the best for each CATH/SCOP PSN group as well as the Astral dataset (electronic supplementary material, figure S1). Consequently, henceforth, of all graphlet approaches, we focus only on OrderedGraphlet-3-4. This result indicates that adding sequence-based node (amino acid) order onto regular (non-ordered) graphlets improves upon the latter alone. This result is in alignment with our past work on unsupervised protein comparison [6]. Unlike in our past unsupervised study, in our current study, graphlet feature normalization does not always improve upon non-normalized features, and sometimes it actually worsens accuracy.

Because OrderedGraphlet-3-4, which uses the default 4 Å distance cutoff, is the best graphlet approach, we also consider the counterpart of OrderedGraphlet-3-4 that uses the 6 Å distance cutoff, i.e. OrderedGraphlet-3-4(6 Å). We find that OrderedGraphlet-3-4(6 Å) is superior to OrderedGraphlet-

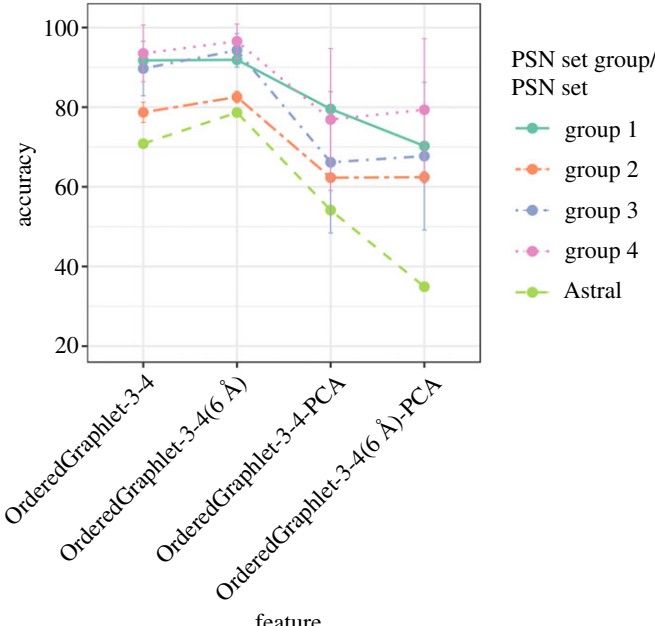

**Figure 2.** Accuracy of the pre- and post-PCA OrderedGraphlet-3-4 (ordered graphlets at the default 4 Å distance cutoff) and OrderedGraphlet-3-4(6 Å) (ordered graphlets at the 6 Å distance cutoff) under the LR classifier, for each of the four hierarchy levels (groups) of the CATH data, averaged over all PSN sets belonging to the given group (vertical lines are standard deviations), plus the Astral PSN set. Results are qualitatively similar for the four groups of the SCOP data as well (electronic supplementary material, figure S2).

3-4 (figure 2). So, we proceed with OrderedGraphlet-3-4(6 Å) as the best graphlet feature. Nonetheless, for informational purpose, we continue to report results for both OrderedGraphlet-3-4 and OrderedGraphlet-3-4(6 Å).

## 3.2. Selection of the best of the pre- or the post-PCA features

Here, we consider the following features under the LR classifier: all non-graphlet features except SVMfold, the two top-performing graphlet features from §3.1 (OrderedGraphlet-3-4 and OrderedGraphlet-3-4(6 Å)), and the two integrated features (i.e. GIT+OrderedGraphlet-3-4(6 Å) and Concatenate) (table 1). Note that since we could not apply SVMfold to all of the 36 PSN sets that we use, we exclude SVMfold from this analysis. When we compare pre- and post-PCA versions of each of the considered features, we find that pre-PCA performs the best for CSM, GIT, OrderedGraphlet-3-4, OrderedGraphlet-3-4(6 Å), GIT+OrderedGraphlet-3-4(6 Å) and Concatenate, while post-PCA performs the best for AAComposition, DAAComposition and Existing-all (electronic supplementary material, figure S3).

So, PCA helps approximately 30% of the time. Thus, henceforth, for each feature, we use the best of its pre- and post-PCA versions. Also, for a given feature, if we use its post-PCA version, '*' is shown next to the feature's name.

## 3.3. Graphlet feature(s) outperform other protein features

First, we compare the best of our PSN-based graphlet features (i.e. OrderedGraphlet-3-4(6 Å)) to a baseline sequence feature (AAComposition) that does not preserve any amino acid order and another baseline sequence feature (DAAComposition) that preserves some amino acid order in a protein sequence. We find that OrderedGraphlet-3-4(6 Å) significantly (adjusted $p$-value, i.e. $q$-value, $< 0.05$) outperforms both AAComposition and DAAComposition in terms of accuracy, but at the expense of a higher (though still more than practical) running time (figures 3 and 4, and electronic supplementary material, figure S4).

Second, we compare OrderedGraphlet-3-4(6 Å) with a state-of-the-art 3D-structural (GIT) feature to see how OrderedGraphlet-3-4(6 Å), which intuitively captures the 3D structure of a protein and is

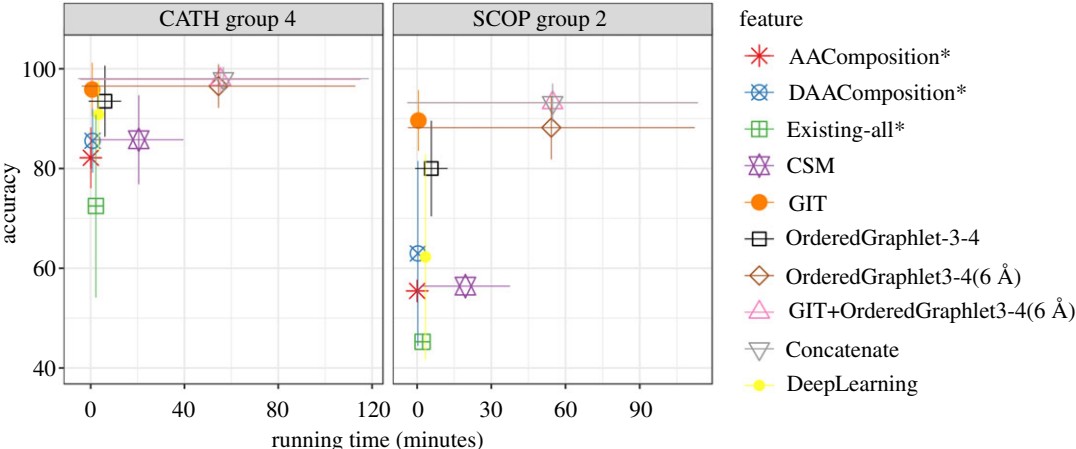

**Figure 3.** Accuracy versus running time of all non-graphlet features except SVMfold, the two top-performing graphlet features from §3.1 (OrderedGraphlet-3-4 and OrderedGraphlet-3-4(6 Å)), and all integrated features (i.e. GIT+OrderedGraphlet-3-4(6 Å) and Concatenate) under the LR classification framework (table 1), for PSN group 4 of CATH and PSN group 2 of SCOP as two representatives. Results are qualitatively similar for all other groups of CATH and SCOP, as well as for the Astral PSN set (electronic supplementary material, figure S4). For each method except DL, the best of its pre- and post-PCA versions is chosen (DL does not have this option). If the latter is selected, '*' is shown next to the given feature's name.

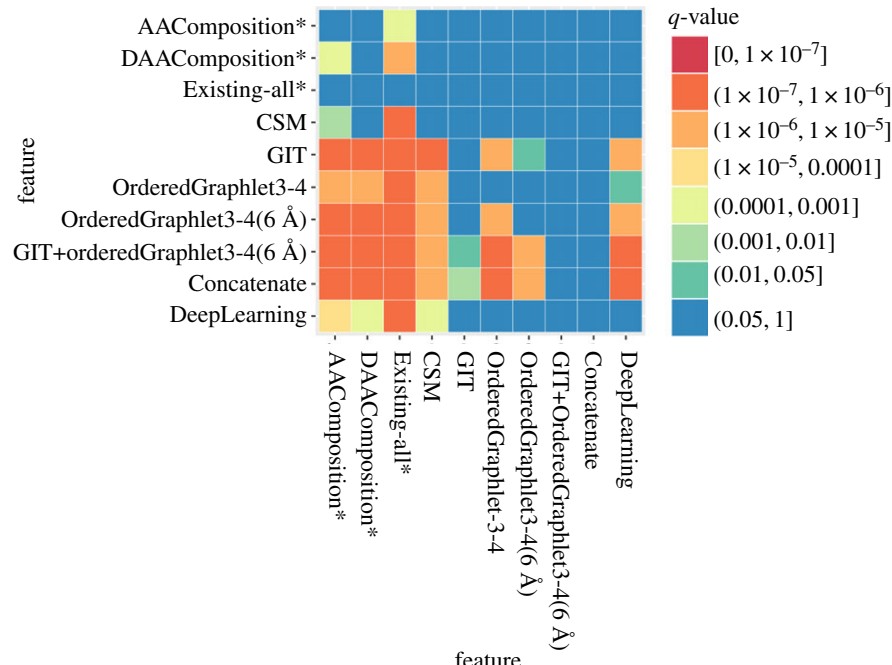

**Figure 4.** Statistical significance of the accuracy difference of the approaches from figure 3. For each of the 36 PSN sets, we measure raw accuracy values for each of the 10 approaches. Hence, for each approach, there are 36 raw accuracy values (corresponding to the 36 PSN sets). For each pair of approaches, we compare the two given approaches' 36 raw accuracy values using Wilcoxon signed-rank test to obtain the corresponding $p$-values. We correct the $p$-values using false discovery rate to obtain the corresponding adjusted $p$-values (i.e. $q$-values). In the figure, every cell $(i, j)$ indicates the statistical significance (in terms of $q$-value) of approach $i$ being superior to approach $j$.

PSN-based, compares against a 3D-structural protein feature that is not PSN-based. We find that on average GIT shows somewhat superior performance in terms of both accuracy and running time over OrderedGraphlet-3-4(6 Å). However, we note the following.

— While GIT shows only somewhat superior performance to the graphlet features, GIT is only applicable to proteins, while graphlets are general-purpose network features and are applicable to many other complex systems that can be modelled as networks.

— Even in the field of modelling protein structures, GIT has the following limitation. GIT is dependent on $\alpha$-carbons to correctly extract 3D-structural feature of a protein and does not process a protein with fewer than 10 or more than 874 $\alpha$-carbons, while graphlet-based features have no such limitation. This is important because according to the current PDB statistics, approximately 11 000 protein chains have fewer than 10 amino acids and approximately 7000 protein chains have more than 874 amino acids. Thus, GIT cannot process any of these approximately 18 000 protein chains, while graphlet-based features can. Note that this is not a problem in our analysis because we only include those protein structures that could be processed by each of the approaches that we consider, including GIT.

— Furthermore, notice that above, we have compared OrderedGraphlet-3-4(6 Å)) against GIT by measuring the performance of each approach as the percentage of correctly classified protein structures over all protein structural classes in a PSN set. By doing so, we lose information on how the given approach performs with respect to each of the structural classes individually (§2.2). So, although *on average* GIT shows somewhat superior performance to the best of our individual graphlet features (i.e. OrderedGraphlet-3-4(6 Å)), it would be interesting to 'zoom into' the results to examine whether OrderedGraphlet-3-4(6 Å) is more suitable for certain protein structural classes compared to GIT. We find that out of all 256 protein structural classes, OrderedGraphlet-3-4(6 Å) outperforms GIT for 84 of the classes (electronic supplementary material, table S4), GIT outperforms OrderedGraphlet-3-4(6 Å) for 132 of the classes, and the two are tied for the remaining 40 classes. That is, OrderedGraphlet-3-4(6 Å) is at least as good as GIT for 124/256 = 48%, i.e. almost half, of all classes. It would be interesting to see whether the three sets of classes (those where OrderedGraphlet-3-4(6 Å) is the best, those where GIT is the best, or those where the two are tied) contain different types of protein secondary structural categories. To examine this, for each class set, we compute the enrichment of its classes in each of the level 1 protein structural categories of CATH and SCOP (which we summarize into '$\alpha$-only', '$\beta$-only', 'both $\alpha$ and $\beta$', and 'other' categories); these categories roughly correspond to different types of secondary structural elements. We find that none of the three sets of classes are significantly enriched in any of the '$\alpha$-only', '$\beta$-only', 'both $\alpha$ and $\beta$', and 'other' categories (electronic supplementary material, section S5). That is, there does not seem to be any secondary structural 'signal' related to which classes are favoured by OrderedGraphlet-3-4(6 Å) versus GIT. Nonetheless, given the complementary performance of OrderedGraphlet-3-4(6 Å) and GIT on all but 40 classes, their integration, i.e. GIT+OrderedGraphlet-3-4(6 Å), might improve upon each of OrderedGraphlet-3-4(6 Å) alone and GIT alone. This is exactly what we observe (see below).

— There is no question that GIT is faster than OrderedGraphlet-3-4(6 Å). Yet, the latter is more than reasonably fast and thus is practically feasible.

Third, we compare OrderedGraphlet-3-4(6 Å) with a recent contact map-based (CSM) feature to see how OrderedGraphlet-3-4(6 Å), which is based on advanced graph-theoretic concepts, compares against CSM, which relies on a 'simple' concept of contact maps. We find that OrderedGraphlet-3-4(6 Å) significantly outperforms CSM ($q$-value < 0.05) in terms of accuracy at the expense of a somewhat higher running time than CSM (figures 3 and 4, and electronic supplementary material, figure S4).

Fourth, we compare OrderedGraphlet-3-4(6 Å) with a baseline network (Existing-all) feature to see how OrderedGraphlet-3-4(6 Å), which is a more comprehensive network measure, compares against Existing-all, which relies on naive network measures. We find that OrderedGraphlet-3-4(6 Å) significantly ($q$-value < 0.05) outperforms Existing-all in terms of accuracy at the expense of a somewhat higher running time than Existing-all (figures 3 and 4, and electronic supplementary material, figure S4).

In summary, these results indicate that OrderedGraphlet-3-4(6 Å) is significantly more accurate than all but one of the considered baseline or state-of-the-art sequence-, contact map-, 3D structure- and PSN-based features. The only feature that is somewhat more accurate than OrderedGraphlet-3-4(6 Å) alone is GIT.

## 3.4. Feature integration improves PSC accuracy

We expect different feature categories (sequence, 3D-structural and PSN-based) to capture different aspects of a protein structure. Thus, integrating features from different categories may help capture complementary protein structural information. Note that we integrate only the best performing feature from each category and do not use other possible combinations of all considered features for two reasons. First, given all of the features that we use (table 1), there are more than 2000 possibilities in which we can combine those features to form a new feature. Hence, evaluating all of the combinations is not feasible. Second, also integrating a poorly performing feature could actually decrease performance compared to integrating only the best feature in each category.

Specifically, first, due to the performance complementary nature of GIT, the best of the 3D-structural features, and OrderedGraphlet-3-4(6 Å), the best of the PSN features (see above), we integrate these two features to form a new feature called GIT+OrderedGraphlet-3-4(6 Å). GIT captures the backbone fold of a protein and OrderedGraphlet-3-4(6 Å) captures both the PSN structure (that captures the 3D structure) and the protein sequence structure using ordered graphlets. Hence, we expect GIT+OrderedGraphlet-3-4(6 Å) to improve upon the two individual features. Indeed, overall, GIT+OrderedGraphlet-3-4(6 Å) significantly improves upon the individual features in terms of PSC accuracy, although at the expense of a higher (yet still more than practical) running time (figures 3 and 4, table 2 and electronic supplementary material, figure S4).

Second, we are also interested in testing whether adding the sequence category to the other two could help further. So, we integrate DAAComposition, the best of sequence-based features that we were able to run on all of our data, with GIT, the best of the 3D-structural features, and with OrderedGraphlet-3-4(6 Å), the best of the PSN features, to form a single feature called Concatenate. Then, we evaluate whether Concatenate improves upon GIT+OrderedGraphlet-3-4(6 Å). We do not see this improvement. Concatenate yields marginal increase in accuracy for some PSN sets but marginal decrease in accuracy for others (table 2). So, overall, adding DAAComposition does not help or hurt the performance of GIT+OrderedGraphlet-3-4(6 Å). Consequently, as expected, just like GIT+OrderedGraphlet-3-4(6 Å), Concatenate shows significant ($q$-value < 0.05) improvement in accuracy compared to each of the three individual protein features, although at the expense of a higher running time (figures 3 and 4, table 2, electronic supplementary material, figure S4 and tables S5–S10).

## 3.5. Robustness of the above results to the choice of training data sampling strategy and performance measure

Recall that we train our LR classification models using two different approaches (proportional or proportional+SMOTE) and evaluate their performances using two different measures (accuracy or MCC) (§2.2), resulting in four different combinations in which we train and evaluate our LR framework. We find that qualitatively, all of the above results remain the same (electronic supplementary material, figures S5–S13 and tables S11–S12). This justifies reporting results for any one of the four strategies throughout the paper. Recall that we have chosen to do so for the combination of proportional sampling and accuracy.

## 3.6. Weighted network-based DL classification performs comparable to unweighted graphlet classification

Our proposed DL classifier performs quite well in terms of accuracy (figure 3 and table 2; electronic supplementary material, figure S4). Specifically, it is significantly ($q$-value < 0.05) superior to AAComposition, DAAComposition, CSM and Existing-all (figure 4). While the accuracy of DL is significantly ($q$-value < 0.05) lower than GIT, OrderedGraphlet3-4, OrderedGraphlet3-4(6 Å), GIT+OrderedGraphlet-3-4(6 Å) and Concatenate (figure 4), DL's accuracy is still pretty high, e.g. over 90% for group 4 of CATH. On average over all 36 PSN sets, DL has accuracy of approximately 81%, which is quite good. As a reference, the best of the individual features has accuracy of approximately 92% and the best of the integrated features has accuracy of approximately 94%.

Importantly, unlike the other individual (LR) classifiers that make use of highly sophisticated unweighted network features such as graphlets, the DL framework uses only as simple as possible weighted network feature (i.e. the weighted adjacency matrix of a network) as its input. This points to a promise of future algorithmic developments of sophisticated features for weighted networks, perhaps even designing weighted graphlet features.

## 3.7. Our features versus SVMfold

SVMfold has a high running time because it needs to extract three sets of very comprehensive features from protein sequence information. This complex information retrieval process needs to be performed for each protein in each considered PSN set, which is not feasible when analysing large PSN sets containing many proteins (such as those at the higher levels of CATH/SCOP hierarchies) or many PSN sets. Hence, we can compare our approaches to the state-of-the-art SVMfold approach only on two representative PSN sets out of all 36 PSN sets. Specifically, for reasons discussed in electronic supplementary material, section S6, we focus on CATH-3.20.20 and CATH-3.40.50 PSN sets. We find

**Table 2.** Accuracy of the approaches from figure 3 for each CATH and SCOP group plus the Astral dataset. Table rows corresponding to our features proposed in this study are coloured in grey, and the rest of the rows correspond to existing approaches. In a column, if any of our proposed features outperforms *each* of the existing features, then the accuracy of the former is shown bold. That is, for all four groups of CATH as well as for two of the SCOP groups, at least one of our proposed approaches is more accurate than all existing approaches.

| approach | CATH | | | | SCOP | | | | Astral |
|---|---|---|---|---|---|---|---|---|---|
| | group 1 | group 2 | group 3 | group 4 | group 1 | group 2 | group 3 | group 4 | |
| AAComposition* | 63.02 | 61.17 | 72.20 | 82.16 | 52.86 | 55.45 | 83.275 | 78.92 | 38.26 |
| DAAComposition* | 69.02 | 64.91 | 79.27 | 85.56 | 57.11 | 62.97 | 83.08 | 87.01 | 40.71 |
| Existing-all* | 82.42 | 62.24 | 60.09 | 72.49 | 63.80 | 45.26 | 69.83 | 74.29 | 30.84 |
| CSM | 85.37 | 69.78 | 74.42 | 85.77 | 72.53 | 56.41 | 80.33 | 87.92 | 43.42 |
| GIT | 91.43 | 81.93 | 95.37 | 95.85 | 81.75 | 89.62 | 91.97 | 97.25 | 81.68 |
| OrderedGraphlet-3-4 | **91.74** | 78.69 | 89.71 | 93.5 | 81.47 | 80.00 | 84.21 | 89.65 | 70.84 |
| OrderedGraphlet-3-4(6 Å) | **91.90** | **82.56** | 94.28 | **96.54** | **82.32** | 88.21 | 88.52 | 95.95 | 78.65 |
| GIT+OrderedGraphlet-3-4(6 Å) | **94.33** | **88.43** | **96.06** | **97.87** | **86.03** | **93.19** | 91.17 | 96.48 | 79.35 |
| Concatenate | **94.20** | **88.42** | **96.10** | **98.03** | **85.93** | **93.23** | 91.50 | 96.48 | 79.48 |
| DeepLearning | 85.72 | **82.94** | 81.63 | 90.85 | 67.67 | 62.27 | 84.36 | 89.96 | 48.84 |

that on these two sets, our best graphlet feature, OrderedGraphlet-3-4(6 Å), and our integrated features, GIT+OrderedGraphlet-3-4(6 Å) and Concatenate, are comparable (within $\pm 5\%$) to SVMfold in terms of accuracy at a fraction of SVMfold's running time (electronic supplementary material, table S13). For more details, see electronic supplementary material, section S6.

## 3.8. The effect of not considering our PSN filtering criteria on the results

Recall that we apply several PSN filtering criteria: we analyse a PSN if and only if it is (i) connected, (ii) has at least 100 nodes, and (iii) has a diameter of at least six (§2.1 and electronic supplementary material, section S1). Here, we examine the effect of these criteria on the relative performance of the considered protein features compared to one another. Specifically, as a proof-of-concept, we analyse a modified Astral PSN set in which we now include a PSN even if it has fewer than 100 nodes or a diameter of less than six (we still focus on connected PSNs only to avoid any bias in our PSC evaluation due to missing data; electronic supplementary material, section S1). The modified Astral set has 2588 PSNs compared to the original Astral set used elsewhere in this paper that has 1677 PSNs (§2.1). We evaluate accuracy of each feature from figure 3 on the modified new Astral PSN set. While we find that the performance of each of the features decreases compared to the performance of the same feature on the original Astral PSN set, the relative performance of the different features compared to each other remains the same as on the original Astral PSN set (electronic supplementary material, figure S14). That is, independent of whether we use our PSN filtering criteria, the results in and conclusions of this paper remain unchanged.

# 4. Conclusion

This study proposes the first ever *network-based* PSC framework. Specifically, this study is the first to use state-of-the-art network-based features called graphlets in the task of PSC. We comprehensively evaluate our graphlet features against state-of-the-art protein features that are based on various other aspects of protein structure, including sequence, 3D structure and contact map information (although we again note some similarity between the notions of a PSN and a contact map). We find that the best of the network-based graphlet features is significantly more accurate than all but one of the considered existing features. Additionally, we show that integrating different protein features improves the PSC accuracy compared to all individual features, possibly by capturing complementary protein structural information. Further, our proposed DL framework, which automatically learns appropriate features from simple weighted PSN adjacency matrices, yields comparable accuracy to many of the sophisticated features that we use, which work on unweighted PSNs. This points to a promising future for algorithms that will rely on weighted network-based features of protein 3D structures, such as weighted graphlets, which currently do not exist.

We show that among all of the considered graphlet features, ordered graphlets perform the best. The superiority of ordered graphlets over the other graphlet and most of the non-graphlet (including 3D-structural and sequence) features might come from the following. First, ordered graphlets comprehensively combine both sequence and 3D-structural information, by incorporating on top of the network patterns that capture the relative ordering of nodes in a PSN, where this ordering relies on sequence positions of amino acids of the corresponding protein. On the other hand, all other features are either sequence- or 3D structure-based but not both. Of course, incorporating not just the relative order of amino acids in the protein sequence but even more of the sequence information into the graphlet approach, such as which particular amino acids appear in which graphlet positions, could yield even further improvements. Developing such an approach is non-trivial and is thus beyond the scope of this study. Second, we believe that studying protein 3D structures as networks is more meaningful than studying the 3D structures directly. This is not just because of the power of networks to reveal interesting data patterns that non-network approaches might miss [61–64], but also because the former allows for a protein structure to be studied with any (current or future) state-of-the-art method for network analysis developed in any field of network science, including the field of protein structural analysis, while the latter allows for using only methods specialized in the field of protein structural analysis. Because the former spans a much larger research community, there probably exist exponentially more network methods that can potentially be used to study protein structures than there exist 3D-structural approaches. In this study, of all current network approaches, we have focused on graphlets, because they have been proven to be the state-of-the-art network

methods, as they, being mathematically sensitive, are able to capture detailed topological information from many different types of complex real-world networks [53].

We evaluate our framework on CATH and SCOP protein domain classes, and in particular on all currently available classes that contain a large enough number of protein domains to have sufficient statistical power in the classification task. As more protein domain data become available, and consequently as more protein domain classes become sufficiently large, our proposed PSC approaches can easily be re-trained on the new data, to allow for classification of protein domains into the new classes as well. Importantly, our framework is a general purpose framework for network classification. That is, although we evaluate our framework in the task of PSC, i.e. when classifying PSNs, our framework can be used to classify networks from any other field, including, but not limited to, other types of biological networks (e.g. protein–protein interaction networks [53]) or social networks (which also have implications for human health [65,66]).

Data accessibility. All data and code are available at https://doi.org/10.5281/zenodo.3787922 [67].

Authors' contributions. Conceived the study: K.N., A.R. and T.M. Collected and processed the data: K.N. Designed the methodology: A.R., M.G. and T.M. Designed the experiments: K.N. A.R., M.G. and T.M. Performed the experiments: K.N., A.R. and M.G. Analysed the results: K.N., A.R., M.G. and T.M. Wrote the paper: K.N., A.R., M.G. and T.M. Read and approved the paper: K.N., A.R., M.G., P.J.A. and T.M. Supervision: P.J.A. and T.M.

Competing interests. The authors have no competing interests.

Funding. This work is funded by the National Institutes of Health (NIH) (grant no. 1R01GM120733). The analyses in this study were partly carried out on the computing infrastructure funded by the National Science Foundation (CNS-1629914).

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
