## [Reviewer comments · Royal Society Open Science]

Review History

RSOS-191461.R0 (Original submission)

Review form: Reviewer 1

Is the manuscript scientifically sound in its present form?

No

Are the interpretations and conclusions justified by the results?

No

Is the language acceptable?

Yes

Do you have any ethical concerns with this paper?

No

Have you any concerns about statistical analyses in this paper?

No

Recommendation?

Major revision is needed (please make suggestions in comments)

Comments to the Author(s)

Summary

Protein structure classification plays a central role in understanding the function of a protein molecule with respect to all known proteins in a database. With the rapid increases in the number of newly captured protein structures (e.g., as deposited in the Protein Data Bank), the development of fast and accurate protein structure classification methods became a key problem in structural bioinformatics.

In the presented paper, the authors focus on the sub-problem of supervised protein structure classification from 3-Dimensional profiles rather than by using the slower, but more usually more accurate protein structure/sequence alignment methods.

In this context, they build upon their previous work on unsupervised classification [Faisal et al., 2016] in which a protein structure is first encoded as a protein Structure Network (PSN, which is the graph representation of the protein contact map, Godzik & Skolnick, 1994) from which a profile is generated using ordered graphlets. Then, using traditional machine learning methods (PCA, Logistic regression), the authors propose NETPCLASS, a supervised protein structure classification software that is more accurate than other supervised protein structure classification methods having comparable running times.

Due to the importance of protein structure classification and the nice preliminary results of NETPCLASS presented in the paper, I am convinced that the paper can become a nice addition to Royal Society Open Science. However, because I have several concerns regarding the method and the experiments, I suggest a major revision.

Major

1: Relatively old competing approaches. In practice, the authors compare NETPCLASS with three profile based supervised protein classification approaches that are relatively old (Pires et al., 2011, and Harder et al., 2012, Xia et al., 2016). The authors should include comparisons with newer methods such as DeepFR (Zhu et al., 2017). Although it is very recent, MV-Fold (Yan et al., 2019) seems to be a tough competitor.

2: Outdated sequence-based profiles. While amino-acid composition profiles (the relative frequency of the 20 amino-acids in the protein sequence) are frequently used in sequence based protein analyses, they have been super-seeded by di- and tri-amino-acid composition profiles, and then by pseudo-amino-acid compositions (PseAAC, Du et al., 2014). All these profiles can be computed by using ProtR/ProtRweb (Xiao et al., 2015).

3: Presented results. I know from reading the previous reviews that testing different distance thresholds for creating PSNs was a last minute addition to the paper. Still, given that the best results are obtained with the newly added threshold of 6Å, this value should be the one used throughout the paper rather than the old 4Å one.

4: Linear machine learning approaches. Graphlets are known to have non-linear redundancies and dependencies (Yaveroglu et al., 2015). Thus, using linear dimensionality reduction (PCA) to transform and reduce the feature space may be ill-suited, which may explain the results from Figures 2 and 3. Using non-linear dimensionality reduction technics may be better suited for this purpose (e.g., principal curve analysis, laplacian eigenmaps, self-organizing maps ...). In the same vein, the proposed predictor is based on Logistic regression, which is also a linear predictor. Again, a non-linear machine learning method may be better suited (e.g., deep neural networks, that the author used but with weighted adjacency matrices).

Minor

5: PSN creation and distance threshold (In line with the comments of past reviewers). There are many ways of defining contacts between amino-acids (distance between alpha carbons, distance between beta carbons, or distance between all heavy atoms, distances between the center of mass of side chains, and even surface based approaches). Furthermore, increasing distance threshold far beyond the usually believed distances of real physical contacts has already been shown to

better perform in CMO based approaches for the task of protein structure classification and comparison (GR-Align was shown to better perform at 12A, and CSM goes up to 30A) [Malodognin et al., 2014, and Pires et al., 2011]. The fact that PSNs look more random when increasing the distance threshold does not mean that they are less informative, but maybe that the signal in them is harder to distinguish from random (also, the randomness was measured using traditional graphlets and not the new ordered ones designed specifically for PSNs [Milenkovic et al., 2009]). Thus, larger distance thresholds and alternative contact definitions could significantly improve the performances of NETPCLASS.

6: Ordered graphlets and their meaning (in line with previous reviewers' comments). While getting a full understanding of why ordered graphlets perform well for protein structure classification, relevant insights could be gained by assessing if amino acids belonging to specific secondary structure elements (alpha, 310, and PI helices, beta strands, parallel and anti-parallel beta sheets, ...) have significant ordered graphlets signatures. Maybe the protein folds that are best predicted by ordered graphlets are rich in those well captured SSEs, while less well predicted protein folds are depleted in those SSEs. (presented results already suggest that protein folds containing mainly alpha helices may be better recognized/predicted than the folds containing beta-strands.)

7. Format issue? The main document is quite long, with about 21 pages including 9 floating elements (tables and figures). I am not familiar with Royal Society Open Science publishing guidelines, but it may be a good idea to reduce the size of the paper by moving less important details and results (e.g., normalized and PCA based features, which are working less well), as well as some tables and figures into the supplemental material. This would also greatly increase the readability of the paper.

Review form: Reviewer 2

Is the manuscript scientifically sound in its present form?

No

Are the interpretations and conclusions justified by the results?

No

Is the language acceptable?

Yes

Do you have any ethical concerns with this paper?

No

Have you any concerns about statistical analyses in this paper?

Yes

Recommendation?

Major revision is needed (please make suggestions in comments)

Comments to the Author(s)

See attached referee report (Appendix A).

Review form: Reviewer 3

Is the manuscript scientifically sound in its present form?

No

Are the interpretations and conclusions justified by the results?

Yes

Is the language acceptable?

Yes

Do you have any ethical concerns with this paper?

No

Have you any concerns about statistical analyses in this paper?

No

Recommendation?

Major revision is needed (please make suggestions in comments)

Comments to the Author(s)

Major:

The authors reason that their method is still valuable even though GIT performs better, since they are the first ones using PSN for classifying protein functionalities. The authors argue that using (weighted) PSN gives rise to new possibilities since it is a new concept and existing network analysis methods can be applied to PSNs.

My issue here is the following: The authors introduce a method for classifying protein functionalities using weighted PSNs and compare their approach to other protein functions classifying methods. The authors argue that the “important” part here is the weighted PSN especially since other network analyses can be applied to study the weighted PSN.

I would prefer to see, that the authors outperform existing methods if the work is focused on a method, or the authors introduce weighted PSN as a concept and show different applications using these PSNs.

- page 3, line 33: The authors say they use supervised learning, in order to outperform existing non-graphlet methods. But this is not true, since GIT is better

- page 4, line 43-45: This work is about predicting protein functionalities and not a general network method. The authors should either introduce a method which can be used for classifying protein functions and which outperforms state-of-the-art methods or introduce a concept, like weighted PSNs and show what can be done with these.

- page 5, line 6-10: The authors claim that their network is not sophisticatedly weighted and that their results would be better if their networks were. How do the authors want to build these networks and why don't they do this in this work?

- page 5, section 2(a): Why do the authors use the two other cutoffs only with their methods but not compare the results to the other methods? It would be interesting to see

- page 10, line 54-59: Does the k-cross validation depends on the k? Did the authors try other values for k?

- page 11, line 16-19: Does the k-cross validation depends on the k? Did the authors try other values for k?
- page 11, line 27-28: But GIT isn't based on PNSs either, right?
- page 13, line 46-47: Their method is outperformed by GIT. So I don't really get the point what is the advantage of their method. Line 49-51: This is not the point of the paper, or? The authors don't introduce the concept of PNSs and show their general applicability ...
- end of page 13, beginning of page 14: their method is better for smaller proteins, less than 10 alpha carbons. But the big ones are the interesting ones or?
- page 14, line 20 ff: the authors do better on a certain type of protein but only 47 out of 256 times. That does not convince me, since it is a very small number.
- page 17, line 9-15: Using weighted networks is not better after all?
- page 17, line 30-34: What would be a more sophisticated way of weighting the matrix? Can the authors give an example and how these matrices can be used? And why don't the authors do this in their work?
- page 17, (f): Since SVM can be compared only using 2 out of the 36 sets, should it be kept in the main text? Maybe in the supplement is sufficient?
- page 18, line 45-56: why don't the authors compare their method using the 6A cutoff to GIT?
- page 19, line 33-39: I don't see why their method points to a promising future when it does not outperform GIT. Maybe the authors can give an example?
- page 19, line 53-57: Not sure if I would agree here. Do methods exist which are solely applied on protein structure prediction? If so, which? Aren't most also applicable to other problems?
- <https://arxiv.org/abs/1203.2821> : graphlets in weighted networks
- Figure 6: GIT+Ordered does not significantly outperform GIT

Minor:

- The writing could be improved. Some sentences are very complex / hard to read. The abstract is not really smooth
- Abbreviations: introduce them when the word is used first (e.g. SS page 2 line 37). Introduce all of them (e.g. SVM, page 2 line 38 or HMM page 2 line 37, page 5 line 17 and 20: Abbrev. For PDB)
- don't introduce abbreviations in titles
- instead of "time complexity" I would use either "running time" or "computational complexity"
- page 2, line 9: remove dot from title
- page 2, line 14-15: I don't understand the sentence "note existing method."
- page 2, line 23-31 and lines 46-57: repetition?
- page 3, line 52 and 53: Twice "propose". Use different word

- page 3, line 56- page 4 line 8: Sentence is hard to understand
- page 4, line 53-56: does a table with running times exist? If so, refer to it. If not, create one
- page 5, line 15: why bold?
- page 8, line 6: typo: "If only if". There is something missing
- page 8, line 35-40: Restructure the sentence. It is not nicely written
- page 8, line 44-49: Restructure the sentence. It is not nicely written
- page 9, line 10-15: Restructure the sentence. It is not nicely written
- page 9, line 18-25: Restructure the sentence. It is not nicely written
- page 9, line 31-34: Restructure the sentence. It is not nicely written
- page 9, line 40-44: Restructure the sentence. It is not nicely written
- page 10, line 8: Underline "NormOrderedGraphlet-3" and "NormOrderedGraphlet-3-4" too for consistency
- page 10, line 38-40: Restructure the sentence. It is not nicely written
- page 10, line 43: dot after ")"
- page 10, line 46: Remove abbrev. Or "linear regression"
- page 11, line 10: No intro of abbrev. In title
- page 11, line 24: Use SVM instead of "support vector machine"
- page 11, line 47: Use DL instead of "deep learning"
- page 18, line 23: "joins" is misleading. I would use "connects"
- page 20, line 6-9: Very vague sentence and not really proven. I would recommend to remove it
- table 1: why is there no vertical line in the last row?
- table 3: Make figure to this table?

Reviewers:

#1

Comment 1: The authors addressed this comment. However, it would be nice to see their method and the different cutoffs compared to the other methods, especially GIT and the other method which outperforms their

#2

I have too little knowledge about this topic to decide if the review's concern is addressed

#3

Comment 11:

I agree to what the authors have done. Not sure however, if their method can be considered as being better. I would say that their method is the first one using PNS and graphlets etc. But their reasoning does not convince me.

The authors say that using a network approach is in general better, since more network analyses methods exist. But then the authors focus on one particular method only. It would be better if the authors introduce their PNSs and use different network analyses methods to show that their new concept is valuable. Or show that their method is better. What the authors do is using one method and one concept which does not outperform existing methods. Their argument is, that their concept is the valuable part. Therefore, the focus of their work should be the concept (PSN), not one particular method

Decision letter (RSOS-191461.R0)

20-Dec-2019

Dear Mr Newaz,

The editors assigned to your paper ("Network-based protein structural classification") have now received comments from reviewers. We would like you to revise your paper in accordance with the referee and Associate Editor suggestions which can be found below (not including confidential reports to the Editor). Please note this decision does not guarantee eventual acceptance.

Please submit a copy of your revised paper before 12-Jan-2020. Please note that the revision deadline will expire at 00.00am on this date. If we do not hear from you within this time then it will be assumed that the paper has been withdrawn. In exceptional circumstances, extensions may be possible if agreed with the Editorial Office in advance. We do not allow multiple rounds of revision so we urge you to make every effort to fully address all of the comments at this stage. If deemed necessary by the Editors, your manuscript will be sent back to one or more of the original reviewers for assessment. If the original reviewers are not available, we may invite new reviewers.

- Data accessibility

<http://datadryad.org/submit?journalID=RSOS&manu=RSOS-191461>

- Competing interests

- Authors' contributions

- Acknowledgements

- Funding statement

Kind regards,
Andrew Dunn

Royal Society Open Science Editorial Office
 Royal Society Open Science
 openscience@royalsociety.org

on behalf of Dr Francois Fages (Associate Editor) and Marta Kwiatkowska (Subject Editor)
 openscience@royalsociety.org

Comments to Author:

Reviewers' Comments to Author:
 Reviewer: 1

Comments to the Author(s)

Summary

Protein structure classification plays a central role in understanding the function of a protein molecule with respect to all known proteins in a database. With the rapid increases in the number of newly captured protein structures (e.g., as deposited in the Protein Data Bank), the development of fast and accurate protein structure classification methods became a key problem in structural bioinformatics.

In the presented paper, the authors focus on the sub-problem of supervised protein structure classification from 3-Dimensional profiles rather than by using the slower, but more usually more accurate protein structure/sequence alignment methods.

In this context, they build upon their previous work on unsupervised classification [Faisal et al., 2016] in which a protein structure is first encoded as a protein Structure Network (PSN, which is the graph representation of the protein contact map, Godzik & Skolnick, 1994) from which a profile is generated using ordered graphlets. Then, using traditional machine learning methods (PCA, Logistic regression), the authors propose NETPCLASS, a supervised protein structure classification software that is more accurate than other supervised protein structure classification methods having comparable running times.

Due to the importance of protein structure classification and the nice preliminary results of NETPCLASS presented in the paper, I am convinced that the paper can become a nice addition to Royal Society Open Science. However, because I have several concerns regarding the method and the experiments, I suggest a major revision.

Major

1: Relatively old competing approaches. In practice, the authors compare NETPCLASS with three profile based supervised protein classification approaches that are relatively old (Pires et al., 2011, and Harder et al., 2012, Xia et al., 2016). The authors should include comparisons with newer methods such as DeepFR (Zhu et al., 2017). Although it is very recent, MV-Fold (Yan et al., 2019) seems to be a tough competitor.

2: Outdated sequence-based profiles. While amino-acid composition profiles (the relative frequency of the 20 amino-acids in the protein sequence) are frequently used in sequence based protein analyses, they have been super-seeded by di- and tri-amino-acid composition profiles, and then by pseudo-amino-acid compositions (PseAAC, Du et al., 2014). All these profiles can be computed by using ProtR/ProtRweb (Xiao et al., 2015).

3: Presented results. I know from reading the previous reviews that testing different distance thresholds for creating PSNs was a last minute addition to the paper. Still, given that the best results are obtained with the newly added threshold of 6A, this value should be the one used throughout the paper rather than the old 4A one.

4: Linear machine learning approaches. Graphlets are known to have non-linear redundancies and dependencies (Yaveroglu et al., 2015). Thus, using linear dimensionality reduction (PCA) to

transform and reduce the feature space may be ill-suited, which may explain the results from Figures 2 and 3. Using non-linear dimensionality reduction technics may be better suited for this purpose (e.g., principal curve analysis, laplacian eigenmaps, self-organizing maps ...). In the same vein, the proposed predictor is based on Logistic regression, which is also a linear predictor. Again, a non-linear machine learning method may be better suited (e.g., deep neural networks, that the author used but with weighted adjacency matrices).

Minor

5: PSN creation and distance threshold (In line with the comments of past reviewers). There are many ways of defining contacts between amino-acids (distance between alpha carbons, distance between beta carbons, or distance between all heavy atoms, distances between the center of mass of side chains, and even surface based approaches). Furthermore, increasing distance threshold far beyond the usually believed distances of real physical contacts has already been shown to better perform in CMO based approaches for the task of protein structure classification and comparison (GR-Align was shown to better perform at 12A, and CSM goes up to 30A) [Malodognin et al., 2014, and Pires et al., 2011]. The fact that PSNs look more random when increasing the distance threshold does not mean that they are less informative, but maybe that the signal in them is harder to distinguish from random (also, the randomness was measured using traditional graphlets and not the new ordered ones designed specifically for PSNs [Milenkovic et al., 2009]). Thus, larger distance thresholds and alternative contact definitions could significantly improve the performances of NETPCLASS.

6: Ordered graphlets and their meaning (in line with previous reviewers' comments). While getting a full understanding of why ordered graphlets perform well for protein structure classification, relevant insights could be gained by assessing if amino acids belonging to specific secondary structure elements (alpha, 310, and PI helices, beta strands, parallel and anti-parallel beta sheets, ...) have significant ordered graphlets signatures. Maybe the protein folds that are best predicted by ordered graphlets are rich in those well captured SSEs, while less well predicted protein folds are depleted in those SSEs. (presented results already suggest that protein folds containing mainly alpha helices may be better recognized/predicted than the folds containing beta-strands.)

7. Format issue? The main document is quite long, with about 21 pages including 9 floating elements (tables and figures). I am not familiar with Royal Society Open Science publishing guidelines, but it may be a good idea to reduce the size of the paper by moving less important details and results (e.g., normalized and PCA based features, which are working less well), as well as some tables and figures into the supplemental material. This would also greatly increase the readability of the paper.

Reviewer: 2

Comments to the Author(s)
See attached referee report.

Reviewer: 3

Comments to the Author(s)

Major:

The authors reason that their method is still valuable even though GIT performs better, since they are the first ones using PSN for classifying protein functionalities. The authors argue that using (weighted) PSN gives rise to new possibilities since it is a new concept and existing network analysis methods can be applied to PSNs.

My issue here is the following: The authors introduce a method for classifying protein functionalities using weighted PSNs and compare their approach to other protein functions

classifying methods. The authors argue that the “important” part here is the weighted PSN especially since other network analyses can be applied to study the weighted PSN.

I would prefer to see, that the authors outperform existing methods if the work is focused on a method, or the authors introduce weighted PSN as a concept and show different applications using these PSNs.

- page 3, line 33: The authors say they use supervised learning, in order to outperform existing non-graphlet methods. But this is not true, since GIT is better
- page 4, line 43-45: This work is about predicting protein functionalities and not a general network method. The authors should either introduce a method which can be used for classifying protein functions and which outperforms state-of-the-art methods or introduce a concept, like weighted PSNs and show what can be done with these.
- page 5, line 6-10: The authors claim that their network is not sophisticatedly weighted and that their results would be better if their networks were. How do the authors want to build these networks and why don't they do this in this work?
- page 5, section 2(a): Why do the authors use the two other cutoffs only with their methods but not compare the results to the other methods? It would be interesting to see
- page 10, line 54-59: Does the k-cross validation depends on the k? Did the authors try other values for k?
- page 11, line 16-19: Does the k-cross validation depends on the k? Did the authors try other values for k?
- page 11, line 27-28: But GIT isn't based on PNSs either, right?
- page 13, line 46-47: Their method is outperformed by GIT. So I don't really get the point what is the advantage of their method. Line 49-51: This is not the point of the paper, or? The authors don't introduce the concept of PNSs and show their general applicability ...
- end of page 13, beginning of page 14: their method is better for smaller proteins, less than 10 alpha carbons. But the big ones are the interesting ones or?
- page 14, line 20 ff: the authors do better on a certain type of protein but only 47 out of 256 times. That does not convince me, since it is a very small number.
- page 17, line 9-15: Using weighted networks is not better after all?
- page 17, line 30-34: What would be a more sophisticated way of weighting the matrix? Can the authors give an example and how these matrices can be used? And why don't the authors do this in their work?
- page 17, (f): Since SVM can be compared only using 2 out of the 36 sets, should it be kept in the main text? Maybe in the supplement is sufficient?
- page 18, line 45-56: why don't the authors compare their method using the 6A cutoff to GIT?
- page 19, line 33-39: I don't see why their method points to a promising future when it does not outperform GIT. Maybe the authors can give an example?
- page 19, line 53-57: Not sure if I would agree here. Do methods exist which are solely applied on protein structure prediction? If so, which? Aren't most also applicable to other problems?
- <https://arxiv.org/abs/1203.2821> : graphlets in weighted networks
- Figure 6: GIT+Ordered does not significantly outperform GIT

Minor:

- The writing could be improved. Some sentences are very complex / hard to read. The abstract is not really smooth
- Abbreviations: introduce them when the word is used first (e.g. SS page 2 line 37). Introduce all of them (e.g. SVM, page 2 line 38 or HMM page 2 line 37, page 5 line 17 and 20: Abbrev. For PDB)
- don't introduce abbreviations in titles
- instead of “time complexity” I would use either “running time” or “computational complexity”
- page 2, line 9: remove dot from title
- page 2, line 14-15: I don't understand the sentence “note existing method.”
- page 2, line 23-31 and lines 46-57: repetition?
- page 3, line 52 and 53: Twice “propose”. Use different word

- page 3, line 56- page 4 line 8: Sentence is hard to understand
 - page 4, line 53-56: does a table with running times exist? If so, refer to it. If not, create one
 - page 5, line 15: why bold?
 - page 8, line 6: typo: "If only if". There is something missing
 - page 8, line 35-40: Restructure the sentence. It is not nicely written
 - page 8, line 44-49: Restructure the sentence. It is not nicely written
 - page 9, line 10-15: Restructure the sentence. It is not nicely written
 - page 9, line 18-25: Restructure the sentence. It is not nicely written
 - page 9, line 31-34: Restructure the sentence. It is not nicely written
 - page 9, line 40-44: Restructure the sentence. It is not nicely written
 - page 10, line 8: Underline "NormOrderedGraphlet-3" and "NormOrderedGraphlet-3-4" too for consistency
 - page 10, line 38-40: Restructure the sentence. It is not nicely written
 - page 10, line 43: dot after ")"
 - page 10, line 46: Remove abbrev. Or "linear regression"
 - page 11, line 10: No intro of abbrev. In title
 - page 11, line 24: Use SVM instead of "support vector machine"
 - page 11, line 47: Use DL instead of "deep learning"
 - page 18, line 23: "joins" is misleading. I would use "connects"
 - page 20, line 6-9: Very vague sentence and not really proven. I would recommend to remove it
- table 1: why is there no vertical line in the last row?
 - table 3: Make figure to this table?

Reviewers:

#1

Comment 1: The authors addressed this comment. However, it would be nice to see their method and the different cutoffs compared to the other methods, especially GIT and the other method which outperforms their

#2

I have too little knowledge about this topic to decide if the review's concern is addressed

#3

Comment 11:

I agree to what the authors have done. Not sure however, if their method can be considered as being better. I would say that their method is the first one using PNS and graphlets etc. But their reasoning does not convince me.

The authors say that using a network approach is in general better, since more network analyses methods exist. But then the authors focus on one particular method only. It would be better if the authors introduce their PNSs and use different network analyses methods to show that their new concept is valuable. Or show that their method is better. What the authors do is using one method and one concept which does not outperform existing methods. Their argument is, that their concept is the valuable part. Therefore, the focus of their work should be the concept (PSN), not one particular method

Author's Response to Decision Letter for (RSOS-191461.R0)

See Appendix B.

RSOS-191461.R1 (Revision)

Review form: Reviewer 1

Is the manuscript scientifically sound in its present form?

Yes

Are the interpretations and conclusions justified by the results?

Yes

Is the language acceptable?

Yes

Do you have any ethical concerns with this paper?

No

Have you any concerns about statistical analyses in this paper?

No

Recommendation?

Accept as is

Comments to the Author(s)

While I understand that the paper was first submitted in December 2018, it is weird that a paper that may be published in 2020 only considers previous works from up to 2017. Due to the complicated revision process of the paper, I'll leave the appreciation of this matter to the Editor. Apart from this, all my previous comments has been answered.

Review form: Reviewer 2

Is the manuscript scientifically sound in its present form?

Yes

Are the interpretations and conclusions justified by the results?

Yes

Is the language acceptable?

Yes

Do you have any ethical concerns with this paper?

No

Have you any concerns about statistical analyses in this paper?

No

Recommendation?

Accept as is

Comments to the Author(s)

I am very happy with all the work done by the authors to address my concerns, including those on the code. I recommend publication as is.

Decision letter (RSOS-191461.R1)

05-May-2020

Dear Mr Newaz,

It is a pleasure to accept your manuscript entitled "Network-based protein structural classification" in its current form for publication in Royal Society Open Science. The comments of the reviewer(s) who reviewed your manuscript are included at the foot of this letter.

In accepting the paper, there are two outstanding tasks you must complete:

1 -- Please ensure that you send to the editorial office an editable version of your accepted manuscript, and individual files for each figure and table included in your manuscript. You can send these in a zip folder if more convenient. Failure to provide these files may delay the processing of your proof.

2 -- You have included your data at your lab website (<https://www3.nd.edu/~cone/NETPCLASS/>); however, we would much prefer that the data are made accessible at a repository such as the OSF (<https://osf.io/>) or via Zenodo (<https://zenodo.org/>) - this has the advantage to you of providing a DOI for your data and code that may be cited. Please can you send me the link to your deposited data as soon as possible?

on behalf of Dr Francois Fages (Associate Editor) and Marta Kwiatkowska (Subject Editor)
openscience@royalsociety.org

Associate Editor Comments to Author (Dr Francois Fages):

Dear Authors

It is my pleasure to finally accept your revised paper as is. You did a good job in addressing the concerns of the reviewers and we understand that the evaluation cannot be further updated.
Best regards

Reviewer comments to Author:

Reviewer: 1

Comments to the Author(s)

While I understand that the paper was first submitted in December 2018, it is weird that a paper that may be published in 2020 only considers previous works from up to 2017. Due to the complicated revision process of the paper, I'll leave the appreciation of this matter to the Editor. Apart from this, all my previous comments has been answered.

Reviewer: 2

Comments to the Author(s)

I am very happy with all the work done by the authors to address my concerns, including those on the code. I recommend publication as is.

Appendix A

Referee report on manuscript RSOS-191461 “Network-based protein structural classification” by K. Newaz, A. Rahnama, M. Ghalehnovi, P. J. Antsaklis and T. Milenkovic.

In their paper Newaz et al. propose to use network properties, graphlets, to train a structure classifier using supervised learning. The authors use both a Logistic Regressor (LR) classifier to compare the resolving power of different graphlet features, comparing them against features already present in literature. The authors further introduce a deep learning framework in order to use more advanced network features.

I believe the authors' claim that topological properties of proteins can be useful in protein structural classification; I also agree that local network properties such as graphlets hold promise. Nonetheless, I do not find the evaluation framework presented by the authors convincing, and I have deep concerns regarding their machine learning procedure and model comparison. As a consequence, I do not think the analyses performed by the authors support their claim regarding the quality of their method, as detailed in this report.

I think the author can address my concerns, which are reported below, and I therefore suggest publication only after major revisions.

Major concerns

Comment 1: use of average accuracy. The authors use the average accuracy to compare different models and decide which one of them gives the best results. This would be OK on balanced datasets, but both CATH and SCOP have several unbalanced classes, including the classes reported by the authors in the supplementary material. Accuracy can give wrong results on unbalanced data, particularly by overestimating the quality of a classifier. An extreme example: one could produce a classifier with ~61% accuracy in CATH 1 (alpha) simply by always choosing class 1.1, orthogonal bundle. The authors mention that the set used for cross-validation has the same proportion of classes, but not that the classes are balanced in any way.

The authors should therefore present more convincing metrics, as e.g. one among cross-entropy, Matthews Correlation Coefficient, Cohen's kappa. The confusion matrices for the various CATH and SCOP groups must also be reported in the supplementary material. All the mentioned metrics are available in Scikit-learn.

Comment 2: use of p-values. I consider the use of p-values, particularly based on a poor metric as reported in comment 1, to be highly problematic in this paper. Not only there exist several metrics (see above) to compare machine learning models, a p-value near 0.05 offers

only weak evidence against the null hypothesis (see e.g. the ASA's statement on p-values: *The American Statistician*, 2016, 70 (2), 129-133).

Comment 3: use of regularization. The effect of regularizers can be different for different features. The authors use just the default value provided by sklearn, both for the regularizer type and strength. A proper study of their effect should be included before reaching a general conclusion. Note: when doing this be careful to keep an extra test set to compare the models after regularizers have been optimized.

Comment 4: assumptions on the test set. Here I might have missed something, but it is not clear to me how does the method perform when the CATH or SCOP classes do not contain at least 30 PSN. If I understood correctly, the authors completely exclude those cases, both from their training set and from their test set. When confronted with a natural protein though, this information is not known, and the protein might well pertain to one of the missing sets (e.g. to "few secondary structures"). The model has no way to know this, and will try to assign the protein to one of the set it knows. So, if this is the case, the reported accuracies are further overestimated: they correspond in fact to training errors and not to test errors, since they are based on the implicit assumption that at least 30 PSN are present. Note that the other methods against which the methodology is compared do not have this specific limitation.

Other comments

Comment 5: missing description of graphlets. Graphlets are barely introduced in the paper, with just a note at page 4, which reference to previous articles. Since they are the main point of the manuscript I believe they are worth of a more detailed introduction. In relation to this, the authors should spend more words to explain what they are and how they can implemented instead of repeating that they are "state of the art". I do not doubt that, but I believe it will be more constructive for the unfamiliar reader to better introduce them than to point out what can be perceived as a subjective statement.

Comment 6: description of set choices (Fig 1). I find the description of how the sets in groups 1 to 4 are performed to be improvable. I think I got the correct idea in the end, but the description should be streamlined.

Comment 7: comparison against GIT (pag 14). At page 14 the authors note that GIT is limited to proteins with 10 alpha-carbons, and states that graphlet-based features have no such limitation. Yet, at page 6 they state that they keep a PSN only if it has at least 100 nodes, i.e. 100 amino-acids. I therefore can not help but find the statement on GIT vs NETPCLASS to be highly misleading.

Comment 8: multiclass classifier (LR): The authors should clarify their choice of how the single class are selected. Why not using softmax, which gives a normalized probability?

Comment 9: PCA transformation. I found the information on PCA transformation to be particularly scant, limiting the reproducibility of the results. The authors should include more details about this, at least in the supplementary.

Comment 10: Feature integration. How do the author perform feature integration, particularly in concatenate-all? Is the class decided according to a majority rule? This should be clarified.

Comment 11: language. I encourage the authors to have a native English speaker proof-reading their manuscript.

Comment 12: python code. I deeply appreciate the fact that the authors published their code. In order to increase its usage I suggest including a properly formatted docstring -preferably using numpy style- in each function and implement pep8 recommendations. Note that some IDEs do the second thing automatically.

Appendix B

The authors' responses to the reviewers' comments

Introduction: general comments to the Editor and all reviewers

We would like to thank the Editor and the reviewers for the opportunity to submit the revised version of our manuscript. We believe that the comments we received, along with our changes in response to the comments, have improved our paper. We believe that we have either explicitly addressed the reviewers' comments or have provided a reasonable justification why we did not address some of them, but we would be happy to make additional modifications if deemed necessary. To summarize, in our effort to thoroughly address each of the reviewers' comments, we have now quadrupled our amount of analyses compared to what we had already done previously. Below, we list each of the reviewers' comments (in italic), and immediately after we provide our corresponding response, along with describing the corresponding change(s) we implemented in the revised paper. (When we list a citation in this response letter, the citation matches that in the bibliography of the main paper.)

Before we continue with our responses to the reviewers' comments, we would like to give a bit of context about the timeline of our paper, i.e., the corresponding study, because this will be relevant to responding to some of the comments. Our study (including evaluation) was designed in 2017 and completed in 2018. Then, we wrote our paper and submitted it to the Journal of the Royal Society Interface in December of 2018. We heard back from the journal in March of 2019, with the invitation to submit to Royal Society Open Science after addressing the reviewers' comments that we got at the time. Here is what the Journal of the Royal Society Interface said to us:

"[Based] on the advice of the reviewers and the editor(s) who handled your manuscript, we would like to offer you the opportunity to transfer your manuscript to another Royal Society journal, Royal Society Open Science... If you opt to have your manuscript transferred to Royal Society Open Science, [ensure] that you revise your text to address all of the reviewers' comments relating to the scientific soundness of the manuscript... Please also ensure that you include a clear point-by-point cover response that details the changes you have made to your manuscript in response to the reviewer(s) and editor(s) who recommended that your paper be transferred. Once submitted to Royal Society Open Science your manuscript will be assessed by an Associate Editor who will decide whether further reviewer advice is required. This may include returning the manuscript to the original referees who assessed the paper at Journal of the Royal Society Interface. If no further advice is needed and all of your revisions are deemed satisfactory, your manuscript will be immediately accepted for publication."

After explicitly addressing all of the reviewers' comments, we submitted our paper to Royal Society Open Science in August 2019, and we heard back with a decision to revise in December 2019.

We have several comments in the light of the above information:

1. We would like to emphasize that we were told by Royal Society Interface that our paper would either be immediately judged by the Associate Editor of Royal Society Open Science (and accepted if all revisions were deemed satisfactory) or that the paper would be returned to the original reviewers. So, you can imagine our shock when we learned that our paper was assigned to entirely new reviewers (which we assume is what happened based on the fact that all comments we received in the latest review process were entirely new). We were not made aware that that our paper could have been sent for another, entirely independent round of review. While we respect that Royal Society Open Science has the ultimate decision on how to handle any paper submitted to that journal, it is frustrating to us (the authors) that what we were told did not match what has happened with our paper when submitted to Royal Society Open Science, as this mismatch has delayed a potential publication of our paper for several additional months. Importantly, this is the issue that we have with the journal(s) and not the reviewers; we greatly appreciate the additional comments

that we have received by the newest set of reviewers, as they have further helped improve the quality of our paper.

2. Because our study (including all of evaluation) was designed in 2017 (i.e., this is when we did a thorough literature search on competing methods) and conducted in early/mid 2018, and because Royal Society (either Interface or Open Science) has held our paper in review for a total of ~8 months, even though we are currently in 2020, we believe that any methods published in 2018 or later can be considered as concurrent to our methods proposed in this paper, and thus, we should not be asked to compare against them or be expected to outperform them.
3. For the same reasons as in item 2 above, it should be no surprise that since our original paper submission in late 2018 we have already completed a follow-up study resulting from the work proposed in this paper and that we have recently submitted the corresponding new paper. Namely, in the current paper (being considered by Royal Society Open Science), we have shown the power of both unweighted graphlet and weighted (adjacency matrix) features in the task of protein structural (and in general, network) classification. So, we concluded that paper with a discussion about a potential of developing weighted graphlets. Consequently, in our follow-up work, which started way after we had submitted the original version of the current paper (being considered by Royal Society Open Science), we have actually developed a notion of weighted graphlets, which further improves classification accuracy. The corresponding paper has just recently been submitted and is thus not published yet (the current version, which we expect will drastically evolve based on comments that we expect to receive from the reviewers, is available on arxiv: <https://arxiv.org/abs/1910.02594>). We believe that the fact that the current paper (being considered by Royal Society Open Science) has motivated this successful follow-up work on weighted graphlets only further justifies the need for publication of the current paper (after the reviewers' judge their comments to be addressed in a satisfactory manner, of course).
4. Related to item 3 above, a reviewer asked us about an existing arxiv paper titled "Graphlet decomposition of a weighted network" (<https://arxiv.org/abs/1203.2821>). However, as the authors of this paper state, their paper is not about graphlets (aka network motifs) as they are typically used in our paper and in general in the field of network science. That is, they say: "We term our method 'graphlet decomposition', as it is reminiscent of a wavelet decomposition of a graph. An unrelated literature uses the term graphlets to denote network motifs (Przulj et al., 2006)", where Przulj et al., 2006 are the ones who proposed graphlets as they are defined in our paper. So, this paper is entirely unrelated to our work, i.e., it is not about graphlets in weighted networks. In fact, our paper work mentioned in item 3 above is the pioneering effort in proposing weighted graphlets.

Thank you all for your understanding.

Now, we proceed onto responding to the individual reviewers' comments.

Reviewer #1

Comment 1: Relatively old competing approaches. In practice, the authors compare NETPCLASS with three profile based supervised protein classification approaches that are relatively old (Pires et al., 2011, and Harder et al., 2012, Xia et al., 2016). The authors should include comparisons with newer methods such as DeepFR (Zhu et al., 2017). Although it is very recent, MV-Fold (Yan et al., 2019) seems to be a tough competitor.

Our response to comment 1: Please see our point 2 above. It is in 2017 when we did a thorough literature search for all competing approaches, the most recent one of which was Xia et al. [59]. While we might have been able to consider DeepFR (see below on why we have not done it even as a part of

this revision), it would be unfair to expect us to compare against a 2019 approach, because our paper was submitted to Royal Society in 2018. Our additional remarks in response to this reviewer’s comment are as follows.

Our study aims to evaluate state-of-the-art protein features in the task of protein structural classification (PSC), when all of the considered features are run under the same classifier. To do this, we have already considered state-of-the-art protein features from several different categories: sequence-based, protein 3-dimensional (3D) structure-based, contact map-based, protein structure network (PSN)-based without using graphlets, or PSN-based with using graphlets. At the time when we started our study (sometime earlier rather than later in 2017), to the best of our knowledge, we used state-of-the-art protein features corresponding to each of the above categories. That is, we used SVM-fold as the state-of-the-art sequence-based protein feature [59], GIT as the state-of-the-art 3D structure-based protein feature [18], contact scanning matrix (CSM) as the state-of-the-art contact map-based protein feature [43], Existing-all as the state-of-the-art PSN-based protein feature that does not use graphlets [10], and ordered graphlets as the state-of-the-art PSN-based protein features that do use graphlets [10].

To the best of our knowledge, GIT, CSM, Existing-all, and ordered graphlets are *still* the state-of-the-art features in their respective categories. In contrast, since 2017, there has been some advancements in the category of *sequence-based* protein features. In particular, there are two such newer methods as suggested by the reviewer, i.e., DeepFR [66] and MV-Fold [61]. However, as explained below, we discovered by reading the DeepFR and the MV-Fold papers that neither the DeepFR nor the MV-Fold method significantly outperforms the SVM-fold sequence-based protein feature that we already consider. So, SVM-fold is still a state-of-the-art sequence-based protein feature.

Regarding MV-fold: We do not evaluate the performance of MV-fold in our study because of the following reasons. First, the MV-fold study compared the MV-fold method against the SVM-fold method (the same method that we already use in our paper) on four different protein datasets: DD, EDD, RDD, and TG [61]. As we can see in Table 1 below (that originates from the MV-fold paper), on average, the differences between the performances of the SVM-fold and the MV-fold methods are negligible (p-value of 0.89 when using the Wilcoxon signed-rank test). Second, both the MV-fold and the SVM-fold methods integrate multiple sequence features to perform protein structural classification. SVM-fold uses three different sequence features, i.e., position-specific scoring matrix (PSSM), three-state secondary structure (SS) profile, and HMM profile of the protein sequence. MV-fold uses four different sequence features – the same three sequence features that are also used in the SVM-fold method, plus some physicochemical information in the form of PseAAC. Because three out of four features used in the MV-fold method are also used in the SVM-fold method, and because computing features for SVM-fold took *significantly* large computational time compared to all other considered features, to the point that we could run SVM-fold on only 5% of our data sets (as explained in the Supplementary Table S13), computing features for MV-fold would definitely be even more time consuming and feasible for even fewer of our data sets. Given this, and given the insignificant differences between the performances of the MV-fold and the SVM-fold methods (as discussed above), we believe that it is not worth running MV-fold on our datasets.

	DD	RDD	EDD	TG	Mean	Std. Dev.
SVM-fold	77.30%	90.00%	94.50%	86.50%	87.08%	7.29%
MV-fold	83.50%	91.70%	94.80%	85.10%	88.78%	5.36%

Table 1: Percentage accuracy of the SVM-fold and the MV-fold methods for the four datasets considered by the MV-fold study. The second last and the last columns of the table shows the average accuracy values and the standard deviations of the two methods over the four datasets.

Regarding DeepFR: We do not evaluate the performance of DeepFR in our framework because of the following reasons. The MV-fold study (which is newer than the DeepFR study) compared the performance of the MV-fold method with the DeepFR method, as well as with the SVM-fold method, per the above discussion. That study found that MV-fold, and thus the comparable SVM-fold, per the above discussion, outperformed DeepFR. Hence, DeepFR is already outperformed by SVM-fold. So, SVM-fold is still among the best sequence-based protein features, which we already use in our paper.

Nonetheless, we added references to the two studies (DeepFR and MV-fold) to our revised paper, along with a brief discussion of the corresponding approaches and a comment on why we do not consider these two approaches (Section 1 (a)).

Comment 2: Outdated sequence-based profiles. While amino-acid composition profiles (the relative frequency of the 20 amino-acids in the protein sequence) are frequently used in sequence based protein analyses, they have been super-seeded by di- and tri-amino-acid composition profiles, and then by pseudo-amino-acid compositions (PseAAC, Du et al., 2014). All these profiles can be computed by using ProtR/ProtRweb (Xiao et al., 2015).

Our response to comment 2: Done. We appreciate the reviewer's comment. In our study, we already used a *state-of-the-art* sequence-based feature called SVM-fold (see our response to comment 1 of reviewer 1). Additionally, we used a *baseline* sequence-based protein feature called AAComposition, in order to evaluate how well the other considered state-of-the-art protein features perform against an *intentionally naive* sequence-based protein feature. In the revised paper, we still include both SVM-fold and AAComposition. Additionally, as suggested by the reviewer, we have now also evaluated the di-amino acid composition, tri-amino acid composition, and PseAAC features. We have found that di-amino acid composition performs significantly better than both tri-amino acid composition and PseAAC. Hence, of all three newly suggested sequence features, for simplicity/aesthetics, we only report results in the paper for the best one, i.e., di-amino acid composition. Importantly, even though this feature performs better than AAComposition, as suspected by the reviewer, it is still inferior to our proposed graphlet approach. So, all results/conclusions of our paper still hold. For the corresponding changes in the revised paper, see paragraph six and seven of Section 2(b), and Section 3(c).

Comment 3: Presented results. I know from reading the previous reviews that testing different distance thresholds for creating PSNs was a last minute addition to the paper. Still, given that the best results are obtained with the newly added threshold of 6Å, this value should be the one used throughout the paper rather than the old 4Å one.

Our response to comment 3: Done. We now add results for the best of our graphlet-based features (i.e., OrderedGraphlet-3-4) that is based on the threshold of 6Å, to the results that compare all considered methods. Because OrderedGraphlet-3-4 with 6Å (i.e., OrderedGraphlet-3-4 (6Å)) significantly outperforms OrderedGraphlet-3-4 with 4Å (i.e., OrderedGraphlet-3-4 (4Å)), we now also update the two of our proposed integrated features that rely on OrderedGraphlet-3-4, i.e., GIT+OrderedGraphlet-3-4 and Concatenate-all, to use the 6Å threshold.

Overall, our conclusions are the same. Our proposed features (either graphlets alone or graphlets integrated with the existing features) still outperform each of the existing considered features. For details, see Sections 3(c) and 3(d).

Comment 4: Linear machine learning approaches. Graphlets are known to have non-linear redundancies and dependencies (Yaveroglu et al., 2015). Thus, using linear dimensionality reduction (PCA) to transform and reduce the feature space may be ill-suited, which may explain the results from Figures 2 and 3. Using non-linear dimensionality reduction technics may be better suited for this

purpose (e.g., principal curve analysis, laplacian eigenmaps, self-organizing maps ...). In the same vein, the proposed predictor is based on Logistic regression, which is also a linear predictor. Again, a non-linear machine learning method may be better suited (e.g., deep neural networks, that the author used but with weighted adjacency matrices).

Our response to comment 4: Regarding redundancies and dependencies in graphlet counts, we have the following response. As per our understanding, Yaveroglu et al. 2014 defined *linear* equations to capture relationships among graphlet counts. So, that study alone does not make a convincing argument that using *non-linear* dimensional reduction techniques would work better than using a linear dimensional reduction technique (e.g., Principal Component Analysis (PCA) in our study). In fact, our research group already showed that PCA works better than two well-known non-linear methods, i.e., minimum curvilinear embedding (MCE) and non-centered MCE (ncMCE), for dimensional reduction of graphlet-based features when comparing biological networks [17]. So, we also proceed with using PCA in our study. We have now added this discussion in Section 2 (b).

Regarding our use of logistic regression (LR) as the machine learning (classification) algorithm in our framework, we have the following response. It would not be surprising for any feature to perform at least somewhat differently under different machine learning algorithms. However, choosing the “best” machine learning algorithm for a given feature is not the aim of this paper. As already explained in the paper, the aim of this work is to evaluate different protein structural features under the *same*, reasonably chosen, machine learning framework, to determine whether graphlets are powerful features in our considered PSC task. By reasonably chosen, we mean a well-performing yet not too computationally complex one. Even the *conventional* LR machine learning algorithm already yields, for the best features, the performance (averaged over the 36 PSN sets) of over ~95%. (As mentioned in the paper, we did initially also consider using a support vector machine (SVM), but LR was still better.) It was already shown that in situations where we already have such well-performing features under a *conventional* machine learning algorithm, *non-conventional* machine learning (e.g., deep learning-based) algorithms might not provide that much of a performance boost compared to the conventional machine learning algorithm (see the reference below). Also, it is well known that using *non-conventional* machine learning algorithms, e.g., deep convolutional neural networks or recurrent neural networks, would come at a cost of much larger running time. So, we believe that in order to evaluate the power of graphlets over the existing PSC features, which is the key task of our paper, there is no need to use more complex (linear or non-linear) classification algorithms on top of the *current* graphlet (and thus any other considered) features, as LR already performs well enough on these features. We have now added this discussion in Section 1 (b). Importantly, if we came up with a deep learning-based machine learning algorithm that when applied to the current graphlet features would significantly improve upon our current results, this would only further strengthen our current study/evaluation/results, which we already believe are strong enough on their own and thus already warrant a publication on their own (after all of the reviewers’ comments have been addressed in a satisfactory manner, of course).

The reason why in our study we do use a deep convolutional network algorithm (and why we do so on top of the weighted adjacency matrix features rather than current graphlet features) is as follows. While PSNs are weighted networks, using current graphlets, which are unweighted, on the weighted PSNs, loses the weight information. So, in our study, we ask whether considering the weights could help. However, unlike for unweighted networks (i.e., unweighted versions of PSNs) for which graphlets are powerful features, there are no established/published features that capture well the topology of a *weighted* network (e.g., a weighted PSN). So, using a weighted PSN adjacency matrix in a deep neural network-based machine learning algorithm that can automatically extract important features from the raw matrix data seemed like an obvious solution to the problem. We have added this discussion in Section 1 (b).

Importantly, the fact that this combination of very simple weighted adjacency matrix feature and deep learning algorithm performed quite well (although not as good as powerful unweighted graphlet features under LR) is what has motivated us to work in 2019 and early 2020 on extending the current unweighted graphlets into weighted graphlets and also to develop a novel deep learning approach for the weighted graphlets, for which we now have preliminary results that the latter improve accuracy further compared to unweighted graphlets under LR. Note that the weighted graphlet study, while motivated by the current study, has been done entirely separately, and the corresponding paper, which is currently under review, thus warrants a separate publication. For details, see our point 3 in the Introduction at the beginning of this response letter.

References:

- Deep learning and alternative learning strategies for retrospective real-world clinical data, Nature Digital medicine, 2, Article number: 43 (2019)

Comment 5: *PSN creation and distance threshold (In line with the comments of past reviewers). There are many ways of defining contacts between amino-acids (distance between alpha carbons, distance between beta carbons, or distance between all heavy atoms, distances between the center of mass of side chains, and even surface based approaches). Furthermore, increasing distance threshold far beyond the usually believed distances of real physical contacts has already been shown to better perform in CMO based approaches for the task of protein structure classification and comparison (GR-Align was shown to better perform at 12A, and CSM goes up to 30A) [Malod-dognin et al., 2014, and Pires et al., 2011]. The fact that PSNs look more random when increasing the distance threshold does not mean that they are less informative, but maybe that the signal in them is harder to distinguish from random (also, the randomness was measured using traditional graphlets and not the new ordered ones designed specifically for PSNs [Milenkovic et al., 2009]). Thus, larger distance thresholds and alternative contact definitions could significantly improve the performances of NETPCLASS.*

Our response to comment 5: We agree that there are several ways to represent an amino acid as a node in a protein structure network (PSN) (or a contact map) and also to define a distance threshold to create an edge between two nodes in a PSN. We do not evaluate each one of such combinations in our study because of the following reasons.

- Using our present choices of node representations and distance cutoffs to create PSNs already yields average accuracy over all considered PSN sets of ~95% for the best approaches. So, investing time, effort, and money to further improve by revisiting the PSN construction is not worth it. Also, a study on the effect of choices of the different PSN construction parameters would be of such a large scope that it might warrant a separate publication on its own. As we argue below, we have done due diligence to choose meaningful values of these parameters.
- To define a node in a PSN corresponding to an amino acid in the given protein, one can use just alpha carbons, just beta carbons, alpha or beta carbons, all heavy atoms, or side chain geometric centers, etc., of the amino acid. In this regard, several past studies evaluated the effect of different node choices in a PSN. However, in general, the conclusions of such studies have been somewhat contradictory. For example, Duarte et al. [9] compared the choice of just alpha carbons, just beta carbons, and alpha or beta carbons, as the three possible choices to define a node in a contact map. They identified that the use of just beta carbon as a node in a contact map captures the 3D protein structure more accurately than the use of just alpha carbon, implying that choice of which atom type to consider for a node in a contact map matters. On the other hand, da Silveira et al. [7] found that the choice of which atom type to consider for a node in a contact map does not make a difference. So, as per our knowledge, there is no consensus on exactly how to choose/define a node in a PSN.

- Even some of the studies that the reviewer has referred to, e.g., the GR-Align study [35], used *only one* of many possible choices of defining a node in a contact map (using alpha carbons to do so). On the other hand, as we already noted in our paper, the GRAFENE study [10] recently showed that whether it was any heavy atom or alpha carbon being used to represent a node in a PSN did not significantly affect results in the task of unsupervised protein structural comparison; no matter the choice, GRAFENE outperformed other state-of-the-art protein structural comparison methods (e.g., DALI [19], TM-align [64], etc.). Because of this, and because we use the same data that was used in the GRAFENE study, we define a node in a PSN in the same manner as the GRAFENE study suggested – using any heavy atom as the choice of a node in a PSN (Section 2(b)).
- Similarly, there is no established criterion for choosing the best distance threshold to use to define whether two amino acids are in contact, i.e., form an edge in the PSN. Several studies tested the effect of different distance thresholds on the meaningfulness of the resulting PSN topologies. However, conclusions of the past studies have been somewhat contradictory. For example, while Vassura et al. [54] showed that an increase in distance cutoff threshold generally increased the meaningfulness of contact maps, Duarte et al. [9] showed that this is not necessarily the case. Additionally, it was shown that the choice of distance threshold often depends on the choice of how a node in a contact map is defined. For example, Yuan et al. [62] showed that for the choice of just alpha carbons as a node in a PSN 8\AA best captures the topology of a protein, for the choice of just beta carbons as a node in a PSN 6.5\AA best captures the topology of a protein, and for the choice of any heavy atom as a node in a PSN 4.5\AA best captures the topology of a protein. Also, Yuan et al. [62] showed that increasing the distance threshold beyond 12\AA actually decreased the performance, irrespective of the choice of node definition of a PSN. So, there is no consensus on which distance cutoff is the best or whether increasing the distance threshold will necessarily increase the meaningfulness of contact maps/PSNs.
- Typically, all of the past contact map/PSN-based studies used those distance cutoffs that gave the best performance for the particular study’s proposed method. For example, as pointed out by the reviewer, *for its own choice of node representation*, GR-Align used a distance cut off of 12\AA as their final distance cutoff. To select this distance cut off, the GR-Align study evaluated the performance of GR-Align itself on a chosen set of distance cut offs, which started from 5\AA and went up to 20\AA , with an increment of 0.5\AA . Then, the study selected the distance cut off for which GR-Align performed the best [35]. In a similar manner, we already evaluated in our originally submitted paper, *for our own choice of node representation*, the best of our graphlet-based features (i.e., OrderedGraphlet-3-4) using three different distance cut offs, i.e., 4\AA , 5\AA , and 6\AA . We start from 4\AA and we go up to 6\AA because, as already pointed out in the paper, it was shown *for the same choice of node representation* that distance cutoffs below 4\AA result in highly disconnected PSNs [38,62], while distance cutoffs beyond 6.5\AA result in random-like PSN structures [38,62]. We are aware of that fact that Milenkovic et al. [38] measured randomness using traditional graphlets and not ordered graphlets. However, currently, that is the best we have in the literature. No measure has been developed yet that uses ordered graphlets to measure randomness in PSNs. Doing that in our current work is out of the scope of this paper.
- Typically, an increase in distance cutoff will also increase the number of contacts/edges in a PSN. Since counting graphlets is a computationally intensive task, higher number of edges typically results in exponential increase in the computational time. Because our currently proposed features that are based on the distance cut off of up to 6\AA already significantly outperform other state-of-the-art features, we believe that increasing the distance cut off further will unnecessarily increase the computational complexity of our proposed PSC framework.

- In general, in our work, the classification accuracy for most of the considered PSN sets increased as we go from 4Å to 6Å. However, we do observe some exceptions. We observe that for two of the PSN sets that we consider in our study, the performance accuracy of OrderedGraphlet-3-4 actually decreases as we go from 4Å to 6Å. Also, for one of the PSN sets the performance accuracy of OrderedGraphlet-3-4 is independent of whether we use 4Å or 6Å. So, we believe that increase or decrease in the performance is somewhat dependent not only on how we define contacts in a PSN, but also on the set of proteins under consideration. It seems that for some PSN sets contacts/edges based on large distance cut offs are important, while for some other PSN sets contacts/edges based on smaller distance cut offs are important. *This is exactly why in our paper we examined whether instead of defining a fixed distance threshold, it would make much more sense to consider the weighted edges.* Hence, we proposed a weighted network-based protein feature, i.e., a PSN's weighted adjacency matrix, which contains pairwise distances between each pair of amino acids of a given protein. We showed that the weighted network-based protein features, combined with our proposed deep learning classifier, achieves accuracy that is relatively comparable to accuracy of the unweighted graphlet features. Note that a weighted adjacency matrix is as simple as possible representation of a weighted network compared to the graphlets that capture a detailed topology of an unweighted network. The fact that weighted adjacency matrix performs relatively comparable to unweighted graphlets, i.e., the outcome of our paper, has opened up an opportunity to develop more advanced representations of weighted networks, such as weighted graphlets (see our point 3 in the Introduction at the beginning of this response letter).
- In summary, we believe that we have selected reasonable and scientifically justified values of the node representation and distance threshold parameters. Requesting us to test all possible values for these parameters, combined with all other requests for other parameters from all reviewers, would yield an exponential number of options to test, which simply is infeasible and thus unrealistic. Again, even our current choice already yields average accuracy over all considered PSN sets of ~95% for the best approaches, so we believe that we have done our due diligence in selecting meaningful values of these and all other parameters in our study. Also, importantly, as the reviewer noted, “testing larger distance thresholds and alternative contact definitions could significantly improve the performances of *NETPCLASS*.” That is, any additional analysis would “only” further strengthen the superiority of *our own* proposed methods, which are already superior to the considered existing methods.

Comment 6: Ordered graphlets and their meaning (in line with previous reviewers' comments). While getting a full understanding of why ordered graphlets perform well for protein structure classification, relevant insights could be gained by assessing if amino acids belonging to specific secondary structure elements (alpha, 310, and PI helices, beta strands, parallel and anti-parallel beta sheets, ...) have significant ordered graphlets signatures. Maybe the protein folds that are best predicted by ordered graphlets are rich in those well captured SSEs, while less well predicted protein folds are depleted in those SSEs. (presented results already suggest that protein folds containing mainly alpha helices may be better recognized/predicted than the folds containing beta-strands.)

Our response to comment 6: We appreciate the reviewer's comment.

In one of our previous studies, which used graphlets for *unsupervised* protein structural comparison [10], we evaluated whether those graphlet patterns that are significantly enriched in “ α only” category of proteins (as defined by CATH/SCOP protein domain annotation databases) are different than those graphlet patterns that are significantly enriched in “ β only” proteins. We found that “ α only” proteins are enriched in denser graphlets (e.g., graphlets where [close to] every pair of nodes

are connected by an edge), while “ β only” proteins are enriched in sparser graphlets (e.g., graphlets that are linear path-like).

Motivated by this, in the current study, we actually already performed a similar analysis in the previous version of our paper that the reviewer has evaluated. Namely, we took all of the CATH/SCOP protein structural classes in which our ordered graphlet-based features performed better than GIT. Then, we computed the enrichment of these classes in each of the level 1 protein structural categories of CATH and SCOP (which we summarize into “ α only”, “ β only”, “both α and β ”, and “other” categories). We found that the classes were significantly enriched in “ β only” (p-value of 0.039). We did a similar analysis for those classes where GIT performed better than OrderedGraphlet-3-4.

Because OrderedGraphlet-3-4 at 6Å (i.e., OrderedGraphlet-3-4 (6Å)) is better than OrderedGraphlet-3-4 at 4Å, we have now updated these results as per OrderedGraphlet-3-4 (6Å). Now, we find that out of all 256 protein structural classes, OrderedGraphlet-3-4 (6Å) outperforms GIT (and all other considered approaches) for 84 of the classes, GIT outperforms OrderedGraphlet-3-4 (6Å) for 132 of the classes, and the two are tied for the remaining 40 classes. (That is, OrderedGraphlet-3-4 (6Å) is at least as good as GIT for $124/256=48\%$, i.e., almost half, of all classes.) For each of the three sets of classes (those where OrderedGraphlet-3-4 (6Å) is the best, those where GIT is the best, or those where the two are tied for the best), we compute the enrichment of the classes in each of the level 1 protein structural categories of CATH and SCOP (which we summarize into “ α only”, “ β only”, “both α and β ”, and “other” categories). Now, we find that none of the three sets of classes are enriched in any of the “ α only”, “ β only”, “both α and β ”, and “other” categories. That is, there does not seem to be any secondary structural signal related to on which classes OrderedGraphlet-3-4 (6Å) vs. GIT performs the best. We have added the new results to Section 3(c).

Hence, our findings have changed compared to the previous paper version, possibly because OrderedGraphlet-3-4 at 6Å is now comparable or superior to GIT on more classes than OrderedGraphlet-3-4 at 4Å was. A more thorough analysis has to be done before understanding the relationship between graphlets and secondary structures and establishing potential generalizable findings. However, given that we have already invested enormous time and effort in the current paper, and given that finding such a relationship is not the goal of study, doing such an analysis is out of the scope of the current paper.

Comment 7: Format issue? The main document is quite long, with about 21 pages including 9 floating elements (tables and figures). I am not familiar with Royal Society Open Science publishing guidelines, but it may be a good idea to reduce the size of the paper by moving less important details and results (e.g., normalized and PCA based features, which are working less well), as well as some tables and figures into the supplemental material. This would also greatly increase the readability of the paper.

Our response to comment 7: Done. We have now shortened some of the text in the paper to the best of our ability. Plus, as suggested by reviewer 3, because we were only able to run SVMfold method on only two of our 36 PSN sets, we moved most of the corresponding discussion and results into the supplement. In total, we have moved two figures and one table to the supplement, now resulting in 6 floating elements left in the main paper. Please note that much of the paper (text, figures, etc.) in the paper is due to already having gone through a paper revision (and now through a second one) and addressing numerous reviewers’ comments, i.e., adding things that were explicitly requested by the reviewers.

Reviewer #2

Comment 1: use of average accuracy. The authors use the average accuracy to compare different model and decide which one of them gives the best results. This would be OK on balanced datasets, but both CATH and SCOP have several unbalanced classes, including the classes reported by the authors in the supplementary material. Accuracy can give wrong results on unbalanced data, particularly by overestimating the quality of a classifier. An extreme example: one could produce a classifier with ~61% accuracy in CATH 1 (alpha) simply by always choosing class 1.1, orthogonal bundle. The authors mention that the set used for cross-validation have the same proportion of classes, but not that the classes are balanced in any way. The authors should therefore present more convincing metrics, as e.g. one among cross-entropy, Matthews Correlation Coefficient, Cohen's kappa.

Our response to comment 1: Done. In addition to accuracy, we now also perform all of our analyses using one of the suggested performance evaluation measures, namely Mathew's Correlation Coefficient (MCC). Additionally, for both accuracy and MCC, in addition to how we handled data originally (via proportional sampling), we now also use a different sampling approach called SMOTE [4], which creates a balanced dataset for training the classifier. We have added the corresponding details in Section 2(b). Importantly, irrespective of whether we use accuracy and proportional sampling, MCC and proportional sampling, accuracy and SMOTE, or MCC and SMOTE, the results and conclusions (e.g., the order of methods in terms of their performance) remain qualitatively the same as in the original paper.

Comment 2: The confusion matrices for the various CATH and SCOP groups must also be reported in the supplementary material. All the mentioned metrics are available in Scikit-learn.

Our response to comment 2: Done. We have added this information in Supplementary Tables S5-S10. Some relevant information:

We consider a total of 35 features and a total of 36 PSN sets. Because there exists a confusion matrix for each feature and PSN set combination (and not for each feature and PSN group combination, as the reviewer thought), the total possible number of confusion matrices is huge. Providing all of the confusion matrices would make no sense, as it would not be possible for any human being to process such amount of information, and also, our Supplementary Materials would be unreasonably large.

For most of our analyses in the paper, starting with Section 3(c) and Figure 3 in the paper, we focus on 10 specific features, e.g., the best-performing features in each of sequence, 3D structural, and PSN categories. However, even for "only" these 10 features, i.e., for all combinations of these 10 features and all 36 PSN sets, the number of confusion matrices is still unreasonably large.

So, we resort to selecting a subset of these 10 features (namely our best individual graphlet feature, our two integrated features, and GIT) as representative features. Also, we resort to selecting a subset of all PSN sets (namely all six PSN sets of CATH group 4) as representative PSN sets. Then, in Supplementary Tables S5-S10, we provide confusion matrices for these feature-PSN set combinations.

We choose the above four features because they are the best-performing of all 35 considered features.

We choose the six PSN sets from group 4 of CATH because of the following reason. For both CATH and SCOP databases, as we go down from hierarchy level (i.e., group) 1 to 4 of the protein domain structural categories, the 3-dimensional structures of protein domains within each hierarchy become more similar to each other. Hence, as we go down the hierarchy, it is more difficult to tell apart two protein domains that are in the same hierarchy but from different protein structural classes. So, PSN sets from hierarchy 4 of CATH/SCOP (i.e., group 4) are the most stringent datasets to test a PSC

approach on. Between CATH group 4 and SCOP group 4, we picked the one with the largest number of PSN sets, i.e., CATH group 4.

Note that, given a feature and a PSN group, it would not make sense to average over all confusion matrices corresponding for all PSN sets in the group to obtain a single confusion matrix for that feature-PSN group combination. This is because a confusion matrix is dependent on different labels/classes present in a PSN set of a group and there is no meaningful way of averaging confusion matrices of two different PSN sets that have different sets of labels/classes. Hence, we cannot report confusion matrices per PSN set group.

Comment 3: use of p-values. I consider the use of p-values, particularly based on a poor metric as reported in comment 1, to be highly problematic in this paper. Not only there exist several metrics (see above) to compare machine learning models, a p-value near 0.05 offers only weak evidence against the null hypothesis (see e.g. the ASA's statement on p-values: *The American Statistician*, 2016, 70 (2), 129-133).

Our response to comment 3: We appreciate the reviewer's concern. First, now we also consider an additional performance measure (see our response to comment 1 of reviewer 2). Second, even in the original paper, we did not solely (or even mainly) rely on p-values; we actually first compared the actual raw accuracy values (averaged over all data sets in the given group) of the different methods/features, and only then we considered p-values of the accuracy differences between the different considered methods. Third, we thank the reviewer for pointing out the article that highlights how the use of p-value can lead to misinterpretation of results, in particular when p-values are taken as a ground-truth or when there is lack of details regarding actual raw results or p-value estimations. Actually, the article suggests adjusting the p-values for multiple correction using False Discovery Rate (FDR) to obtain the corresponding q-values, and then using the q-values as an alternative measure to compare methods' performance. This is exactly why we have now added FDR analysis to our evaluation in the revised paper. We now consider one approach to be significantly better than another approach if the q-value of their difference is less than 0.05. This is a much more statistically stringent condition than the original condition of unadjusted p-values. For details, see Section 3(d) and Figure 4.

Note that we already provided several figures in the Supplementary file (Figs. S1 to S4) indicating average performances of the considered features over different groups of data sets. However, we did not explicitly provide all of the raw performance values for the considered features. We have now included this information in the Supplementary Tables S11-S12 for full transparency.

Comment 4: use of regularization. The effect of regularizers can be different for different features. The authors use just the default value provided by sklearn, both for the regularizer type and strength. A proper study of their effect should be included before reaching a general conclusion. Note: when doing this be careful to keep an extra test set to compare the models after regularizers have been optimized.

Our response to comment 4: Done.

Regarding regularizer strength: We now vary regularizer strength for a given feature and choose the best-performing value, before evaluating the performance of the feature on separate (yet unseen) test data. Overall, our conclusions are the same as in the previous paper version. Namely, our proposed features (either graphlets alone or graphlets integrated with the existing features) still outperform each of the existing considered features. For details, see Section 2(b) and Supplementary Section S3.

Regarding regularizer type: There is not really a consensus on which of the two regularization is the "best". Existing machine learning studies typically chose only one regularization type, most often arbitrarily. In particular, in existing logistic regression-based machine learning literature, either L1 or L2 regularization is used. Following this common practice, we made a choice to use L2 or Ridge

regularization for our analysis. Our choice is mostly motivated by the fact that L1 is typically used as a feature selection method, while L2 is typically used as a general purpose regularization approach. Since we do not aim for feature selection, we choose L2 regularization in our study.

Comment 5: assumptions on the test set. Here I might have missed something, but it is not clear to me how does the method perform when the CATH or SCOP classes do not contain at least 30 PSN. If I understood correctly, the authors completely exclude those cases, both from their training set and from their test set. When confronted with a natural protein though, this information is not known, and the protein might well pertain to one of the missing sets (e.g. to “few secondary structures”). The model has no way to know this, and will try to assign the protein to one of the set it knows. So, if this is the case, the reported accuracies are further overestimated: they correspond in fact to training errors and not to test errors, since they are based on the implicit assumption that at least 30 PSN are present. Note that the other methods against which the methodology is compared do not have this specific limitation.

Our response to comment 5: Yes, we completely exclude those cases. However, this is not specific to any method that we consider or even to our study, and in fact, the same needs to be done *for all PSC methods and in any machine learning study*, because one needs to ensure that there is a sufficient number of samples in each class, so that there is sufficient statistical power in the classification task. That is, removal of small classes is not a limitation of our proposed approach but a *broadly accepted* precaution to avoid underfitting when training a machine learning algorithm [11].

In a bit more detail, as a part of the evaluation, *as is typically done in any machine learning study*, we hide the CATH/SCOP class knowledge for a subset of the considered proteins, and then we try to recover the classes of these proteins. That is, in the process of evaluation, we use some part of the data to train a classification model and use the remaining part of the data to test the classification model (Section 2(b)). The same class is split between the training and testing data, for each considered class. So, any protein in the testing data can in theory be predicted as belonging to its correct class, because the same class exists in the training data too. If a protein of an unknown class (or a class not in the training data) is given to the classifier, then the classifier will predict for that protein whichever class of those classes considered in the training data the protein is the most likely to belong to. *This is how machine learning works; this has nothing to do with our study per se.* As more protein domain data become available, and consequently as more protein domain classes become sufficiently large, our proposed PSC approaches can easily be re-trained on the new data, to allow for classification of protein domains into the new classes as well. We already had (and still have) a discussion on this in paragraph three of Conclusion.

Comment 6: missing description of graphlets. Graphlets are barely introduced in the paper, with just a note at page 4, which reference to previous articles. Since they are the main point of the manuscript I believe they are worth of a more detailed introduction. In relation to this, the authors should spend more words to explain what they are and how they can implemented instead of repeating that they are “state of the art”. I do not doubt that, but I believe it will be more constructive for the unfamiliar reader to better introduce them than to point out what can be perceived as a subjective statement.

Our response to comment 6: Done. We have now added a description of graphlets in Section 2(b).

Comment 7: description of set choices (Fig 1). I find the description of how the sets in m groups 1 to 4 are performed to be improvable. I think I got the correct idea in the end, but the description should be streamlined.

Our response to comment 7: Done. We have now updated this description in Fig.1 to make it more clear.

Comment 8: comparison against GIT (pag 14). At page 14 the authors note that GIT is limited to proteins with 10 alpha-carbons, and states that graphlet-based features have no such limitation. Yet, at page 6 they state that they keep a PSN only if it has at least 100 nodes, i.e. 100 amino-acids. I therefore can not help but find the statement on GIT vs NETPCLASS to be highly misleading.

Our response to comment 8: Our statement in question *is correct*. GIT certainly cannot process any protein with fewer than 10 alpha carbons. However, our NETPCLASS framework (i.e., our graphlet approach) has no such limitation. This is important because, according to the current statistics of the Protein Data Bank (PDB), ~11,000 protein chains have fewer than 10 amino acids. Thus, GIT cannot process any of these ~11,000 protein chains, while NETPCLASS can.

Additionally, we also found that GIT is not able to process any protein which has more than 874 amino acids, while NETPCLASS has no such limitation. According to the current statistics of the Protein Data Bank (PDB), ~7,000 protein chains have more than 874 amino acids. Thus, GIT cannot process any of these ~7,000 protein chains, while NETPCLASS can.

Note that none of these limitations are mentioned in the GIT paper [18,46] or in the documentation of the GIT software. All of these GIT limitations can only be known after one runs GIT on proteins of different lengths. So, it is important that this information is known, and we state this information in Section 3 (c).

The reviewer is correct that in our original study we did not consider proteins that had fewer than 100 nodes. However, this was not a methodological limitation; instead, it was an evaluation choice – graphlets simply might not be needed for smaller networks. Nevertheless, in the previously revised version of the paper that the reviewer already handled, we already performed an analysis on all proteins, *including those with fewer than 100 nodes* (but still more than 10 and or fewer than 874, in order to match the data that GIT can, i.e., cannot, process); see Section 3 (h). We showed that the relative performance of the different considered methods remained the same irrespective of whether we included or excluded proteins with fewer than 100 nodes, with graphlets being ranked on top.

Comment 9: multiclass classifier (LR): The authors should clarify their choice of how the single class are selected. Why not using softmax, which gives a normalized probability?

Our response to comment 9: As stated in Supplementary Section S3, we select a class in our multi-class PSC problem as follows. We use the one-vs-rest scheme to the LR classifiers. That is, given a PSN set with m protein structural classes, we train m different binary LR models that correspond to the m different classes. Given a test protein, an LR model outputs a score for the corresponding class. Given output scores for m classes, we consider the class with highest output score as the predicted class for the given input protein.

We understand that a softmax regression might have been a better choice than multiple binary LR classifiers. However, as already explained Section 1 (b), choosing the “best” possible machine learning algorithm is not the aim of this paper. Instead, its aim is to evaluate different protein structural features under the *same*, reasonably chosen, machine learning framework, to determine whether graphlets are powerful features in our considered task of protein structural classification (PSC). For additional details on what we mean by “reasonably chosen”, see our response to comment 4 of reviewer 1.

Comment 10: PCA transformation. I found the information on PCA transformation to be particularly scant, limiting the reproducibility of the results. The authors should include more details about this, at least in the supplementary.

Our response to comment 10: Done. We have now added a detailed description of how we performed PCA transformation for each of the PSN sets in Section 2(b).

Comment 11: Feature integration. How do the author perform feature integration, particularly in concatenate-all? Is the class decided according to a majority rule? This should be clarified.

Our response to comment 11: As stated in Section 2 (b) of the paper, in order to obtain our integrated features, we concatenated the respective participating individual features into a single feature. For example, to obtain the integrated feature GIT+OrderedGraphlet-3-4, we concatenated (appended together) the GIT and the OrderedGraphlet-3-4 features of a protein into a single GIT+OrderedGraphlet-3-4 feature for the protein. Then, we use this integrated feature in the same way as we use any other considered individual features to learn a machine learning model for protein structure classification.

Comment 12: language. I encourage the authors to have a native English speaker proof-reading their manuscript.

Our response to comment 12: Done. We hope that the entire paper now reads clearly. But if the reviewer still has any specific follow-up comments on any parts of the text that might remain unclear, we would be happy to address such comments.

Comment 13: python code. I deeply appreciate the fact that the authors published their code. In order to increase its usage I suggest including a properly formatted docstring-preferably using numpy style- in each function and implement pep8 recommendations. Note that some IDEs do the second thing automatically.

Our response to comment 13: Done. Updated code has now been posted on the project website that is referenced in the abstract of the paper.

Reviewer #3

Comment 1: The authors reason that their method is still valuable even though GIT performs better, since they are the first ones using PSN for classifying protein functionalities. The authors argue that using (weighted) PSN gives rise to new possibilities since it is a new concept and existing network analysis methods can be applied to PSNs. My issue here is the following: The authors introduce a method for classifying protein functionalities using weighted PSNs and compare their approach to other protein functions classifying methods. The authors argue that the “important” part here is the weighted PSN especially since other network analyses can be applied to study the weighted PSN. I would prefer to see, that the authors outperform existing methods if the work is focused on a method, or the authors introduce weighted PSN as a concept and show different applications using these PSNs.

Our response to comment 1: First, a minor clarification: Our work is *not* on predicting protein *functionalities*, i.e., it does not deal with the task of protein *functional* classification. Instead, it deals with protein *structure* classification (*PSC*). Of course, structure classification can be further adapted to

function classification, which is what we use to motivate the importance of the task. However, our paper only deals with PSC.

Second, let us clarify: The major goal our paper is to evaluate how well *general* network features, such as (currently *unweighted*) graphlets, when applied to PSN representations of protein structures, perform compared to *PSC-task-specialized* state-of-the-art protein features under the same classifier. If an approach based on (currently unweighted) graphlets is superior in the PSC task to the state-of-the-art existing methods for the same task, then we will have achieved *the key goal of our study – demonstrating the power of PSN-based analyses of protein structures*. Indeed, we find that graphlets, either individually or when combined with existing methods (which is a novel idea that we propose in this paper) outperform all considered existing approaches.

Because current graphlets are unweighted and can thus deal only with unweighted versions of PSNs, which are in reality weighted, the minor goal of our paper is to evaluate whether accounting for edge weights in the PSNs helps. Because currently there exist no effective methods for extracting features in weighted PSNs, the best we could do is to analyze weighted PSN adjacency matrix features of the PSNs. If this weighted approach, which is based on as naïve weighted PSN features as possible, is already superior to or at least achieves reasonably high accuracy compared to existing (and our unweighted graphlet) approaches, then we will have shown the usefulness of considering edge weights in *future*, more powerful weighted network features. Indeed, we find that even the current weighted network approach achieves reasonably high PSC accuracy; being naïve, it is unsurprising that this approach is not the best.

The two above results combined – graphlets, state-of-the-art unweighted network features, as well as naïve weighted network features, working very well in the PSC task, have motivated the need for weighted graphlets. After we submitted this paper to Royal Society in 2018, we have developed weighted graphlets, and indeed, they further improve PSC accuracy. This only further emphasizes the importance of the current paper under consideration in Royal Society, as it motivated the development of weighted graphlets in the first place. For reasons discussed in our point 3 in the Introduction (pages 1-2) of this response letter, the weighted graphlets paper is out of the scope of the current paper, as each study warrants a separate publication.

Bottom line: In the current paper under consideration in Royal Science, we *do* propose novel features that outperform *each* of the considered state-of-the-art protein features in the task of PSC.

Comment 2: page 3, line 33: *The authors say they use supervised learning, in order to outperform existing non-graphlet methods. But this is not true, since GIT is better*

Our response to comment 2: To remind the reviewer, page 3, line 33, of our submitted paper said the following:

“In this work, we use the graphlet-based PSN features for the first time in the task of supervised PSC, with a hypothesis that they will improve upon state-of-the-art non-graphlet PSN features and non-PSN features that have traditionally been used in this task.”

First, a minor clarification: We do not “use supervised learning, in order to outperform existing non-graphlet methods”. Instead, we compare graphlets against the existing non-graphlet methods in a specific task of PSC, which just happens to be supervised. Second, up to this point in the paper, we are just putting forth a *hypothesis* that we are testing – whether our graphlet-based methods will improve upon current non-graphlet-based PSC methods. So, we still stand by this text and thus keep it as is. Third, as we show much later in the paper, in the Results section, our proposed integrated features, which include graphlets, *do* outperform all existing non-graphlet-based features, including GIT.

Comment 3: page 4, line 43-45: This work is about predicting protein functionalities and not a general network method. The authors should either introduce a method which can be used for classifying

protein functions and which outperforms state-of-the-art methods or introduce a concept, like weighted PSNs and show what can be done with these.

Our response to comment 3: For details, see our response to comment 1 of reviewer 1. But briefly, again, first we have a minor clarification: This work is not about predicting *functionalities*, i.e., it is not about protein *functional* classification. Instead, it is about protein *structural* (and in general, network) classification. Second, we *do* introduce a method that classifies protein structures significantly more accurately than all considered state-of-the-art methods for the same purpose. Specifically, our integrated features, which include our proposed graphlets, are the best of all considered existing approaches; even graphlets alone are better than all but one of all considered existing approaches. Third, our graphlets are a general network science approach, and as such, they can be used to study not just protein structures but also protein-protein interaction networks, brain networks, social networks, infrastructure networks, etc., while the specialized protein structural approaches can be used to study only protein structures (see our response to comment 17 of reviewer 3 for details).

Comment 4: page 5, line 6-10: The authors claim that their network is not sophisticatedly weighted and that their results would be better if their networks were. How do the authors want to build these networks and why don't they do this in this work?

Our response to comment 4: To remind the reviewer, on page 5, line 6-10, of our submitted paper said the following:

“Note that here we are comparing as simple as possible weighted network information (the weighted adjacency matrix) against highly sophisticated unweighted network information (graphlet features, which are the state-of-the-art in network science). So, a comparable accuracy of the former and the latter implies a promise of future weighted network-based analyses of protein 3D structures (such as developing and using weighted graphlet-based features).”

We believe that there is some confusion here, so let us try to clarify. There is no approach per se to weighting the networks, i.e., PSNs. The edge weights in a PSN are determined by the protein's 3D structure. What we refer to as “naïve” (i.e., not sophisticated) versus sophisticated is *how we extract features* from a weighted PSN. Namely, on the other hand, current graphlets are state-of-the-art (i.e., highly sophisticated) *unweighted* network features, which we apply to unweighted versions of the weighted PSNs. To evaluate whether accounting for edge weights in the PSNs when extracting features, we could not rely on existing network features, because they do not allow for accounting for edge weights. So, the best we could do is to consider naïve weighted adjacency matrix features of the PSNs. Even such naïve features already yielded reasonably high PSC accuracy. (In our follow-up work, we develop novel weighted graphlet features to further improve accuracy compared to each of sophisticated unweighted graphlets and naïve weighted adjacency matrix features.) For additional information, see our response to comment 1 of reviewer 3. We believe that the reviewer's comment originated from us using somewhat ambiguous terminology of sophisticated vs. simple/naïve sophisticated unweighted network *information*, whereas we meant sophisticated vs. simple/naïve sophisticated unweighted network *features*. We have modified the text in question to update this ambiguity (see paragraph one in section 1 (b) in the revised paper).

Comment 5: page 5, section 2(a): Why do the authors use the two other cutoffs only with their methods but not compare the results to the other methods? It would be interesting to see

Our response to comment 5: The cutoffs mentioned by the reviewer are used to define whether there should be an edge between two amino acid nodes *in a PSN*. So, *only PSN-based approaches are*

affected by this parameter. In other words, this parameter does *not* affect any considered sequence- or 3D structure-based approaches.

As discussed in Section 3(a) of the paper, the best of all PSN methods (when run at the default 4Å distance cutoff) is our OrderedGraphlet-3-4 method. So, for this best PSN method, we examine the effect of the choice of distance cutoff, by varying this parameter from 4Å to 6Å. The latter (using OrderedGraphlet-3-4 at 6Å) performs the best. This result highlighted the fact that by varying the cutoff, our PSN method could only further improve upon existing methods, whose performance would be unaffected by varying this parameter. Please refer to our response to comment 5 of reviewer 1 for a detailed discussion on the selection of a distance cutoff.

Because OrderedGraphlet-3-4 at 6Å is better than OrderedGraphlet-3-4 at 4Å, while in the previous paper, our integrated features considered the latter, in our revised paper, our integrated features consider the former. That is, in the light of the reviewer's comment, we now integrate our OrderedGraphlet-3-4 feature at 6Å with other considered features, which yields a statistically significant superiority of the new integrated features compared to all of the considered existing features. For details, see Section 3(d).

Comment 6: page 10, line 54-59: Does the k-cross validation depends on the k? Did the authors try other values for k?

Our response to comment 6: We do not use other values of k because of the following reason. Evaluation of each and every possible value of k ($k = 2, \dots, N$, where N is the size of a dataset), would require an extensive study. This is because we use 36 different datasets, each with a different size/number of data points. If we consider M to be the smallest number of data points in any of our considered datasets, then in order to perform our study to choose the "best" value for k, we would need to try *at least* $36 * M$ experiments per protein feature. In our considered datasets, $M=64$, and hence, this would result in at least $36 * 64 = 2,304$ experiments per protein feature. Additionally, since we evaluate performances of 32 different protein features, this would mean that we would need to perform at least $36 * 64 * 10 = 23,040$ experiments in total. Therefore, instead of blindly evaluating each and every value of k for our experiments, we trusted in previous research that deal with the determination of the "best" value for k in k-fold cross validation approach. A number of past studies have shown that the value of $k=10$ works quite well in terms of evaluating performances of different machine learning models/algorithms/features [27,50]. Hence, we use $k=10$ for our study. We do note that while working on a different (weighted graphlet) paper that uses the same data in the same task, a new student did use 5-fold cross validation, and we have informally compared those results against the results of 10-fold cross validation for the same method. The results were indistinguishable.

Comment 7: page 11, line 16-19: Does the k-cross validation depends on the k? Did the authors try other values for k?

Our response to comment 7: See our response to comment 6 of reviewer 3.

Comment 8: page 11, line 27-28: But GIT isn't based on PNSs either, right?

Our response to comment 8: To remind the reviewer, page 11, the paragraph containing the lines 27-28, of our submitted paper said the following:

"Given these structural information, the method uses a deep convolutional neural network framework along with support vector machine to perform the protein function prediction. The method was applied to classify enzymes (i.e., proteins) into functional categories. Since we only became aware of this very recent method towards the end of our work, we could not include it

into our evaluation. However, note that this method is not based on PSNs and the focus of the study was on the task of protein function prediction rather than on our task of PSC.”

True, GIT is not based on PSNs either. However, the purpose of this paragraph was to highlight the reasons why we were unable to use the *deep learning* method in question as a benchmark against our proposed weighted PSN *deep learning* approach. If this approach also allowed for analysis of weighted PSNs, we would have certainly considered it. Also, we were mostly being courteous to the authors of that paper/method, as it deserves to be mentioned in our paper. However, because (i) we learned of this approach very late in our evaluation, (ii) the method in question extracts 3D structural features rather than PSN-based features and we already compare to state-of-the-art 3D structural features, including GIT, and (iii) the method was developed to do functional classification of enzymes rather than structural classification of proteins, and we focus on the latter, we did not see value in investing additional time/effort/money into adding yet another approach into our already comprehensive evaluation. Note that unlike this method in question, GIT was exclusively developed for the task of protein structural comparison and is thus much more related to our work. We modified the text in question to include the above discussion and clarifications (see the last paragraph of Section 2 (b) (ii) in the revised paper).

Comment 9: page 13, line 46-47: Their method is outperformed by GIT. So I don't really get the point what is the advantage of their method. Line 49-51: This is not the point of the paper, or? The authors don't introduce the concept of PNSs and show their general applicability ...

Our response to comment 9: For a thorough argument, see our response to comment 1 of reviewer 3. Briefly, first, our graphlet method alone, which is a *general* network approach, performs in the task of PSC very close to GIT, which is a *PSC-specialized* approach. This performance of graphlets is impressive in our eyes, even though GIT is somewhat better than graphlets alone; the somewhat better performance of GIT is not surprising, because this is a PSC-specialized approach. Second, and more importantly, we *do* propose new approaches, namely the integrated features that incorporate our proposed graphlet approach, which *do* significantly outperform GIT (as well as all other considered existing approaches). For details, see Section 3(d).

Comment 10: end of page 13, beginning of page 14: their method is better for smaller proteins, less than 10 alpha carbons. But the big ones are the interesting ones or?

Our response to comment 10: To remind the reviewer, the text lines pointed out by the reviewer are as follows:

*“Even in the field of modeling protein structures, GIT has the following limitation. GIT is dependent on α -carbons to correctly extract 3D structural feature of a protein and does not process a protein with fewer than 10 α carbons, while graphlet-based features have no such limitation. Note that this is not a problem in our analysis because we only include those protein structures that could be processed by each of the approaches that we consider. **Nonetheless, since GIT is dependent on α -carbons, it would be interesting to see how noise-tolerant GIT is to the removal of α -carbons, compared to the best of our graphlet approaches (i.e., OrderedGraphlet-3-4).**”*

The black part of the paragraph pointed out by the reviewer is saying that GIT has a limitation that it cannot process or extract features for any protein that has fewer than 10 α -carbons, and that our graphlet-based approaches have no such limitation. The importance of this drawback of GIT is that according to the current statistics of the Protein Data Bank (PDB), ~11,000 protein chains have fewer than 10 amino acids. Thus, GIT cannot process any of these ~11,000 protein chains, while our graphlet methods can. This is not an issue in our study per se, because as the above text states, when we

evaluated the performance of the considered features, we made sure to include only those proteins that could be processed by each of the considered features, including GIT.

An additional note: We also found that GIT is not able to process any protein which has more than 874 amino acids, while NETPCLASS has no such limitation. According to the current statistics of the Protein Data Bank (PDB), ~7,000 protein chains have more than 874 amino acids. Thus, GIT cannot process any of these ~7,000 protein chains, while NETPCLASS can.

None of the two GIT limitations are mentioned in the GIT paper [18,46] or in the documentation of the GIT software. These GIT limitations can only be known after one runs GIT on proteins of different lengths. So, it is important that this information is known. Consequently, we have modified the above text pointed by the reviewer to include this information (see Section 3 (c) in the revised paper).

Note that we have removed the blue part of the paragraph pointed out by the reviewer and the corresponding noise-tolerance analysis, as it simply did not add anything to our paper and was thus only unnecessarily increasing its length.

Comment 11: page 14, line 20 ff: the authors do better on a certain type of protein but only 47 out of 256 times. That does not convince me, since it is a very small number.

Our response to comment 11: To remind the reviewer, page 14, paragraph starting line 20, in the submitted paper said the following:

“Furthermore, notice that given a PSN set, we measure the performance of a given approach as the percentage of correctly classified protein structures over all of the protein structural classes, without looking into how the given approach performs with respect to each of the structural classes individually (Section 2(b)). So, although on average GIT shows comparable performance to our individual graphlet features, it would be interesting to see whether our individual graphlet features are more suitable for certain protein structural classes as compared to GIT. We find that of all 256 protein classes over all 36 PSN sets, the best of our graphlet approaches, OrderedGraphlet-3-4 and NormOrderedGraphlet-3-4, show better performance than GIT in 28 and 19 protein classes, respectively, i.e., in a total of 47 classes (Supplementary Tables S4 and S5)....”*

First, the purpose of this paragraph was to show that even though overall our *individual* graphlet-based protein features are not significantly outperforming GIT, our individual graphlet-based protein features are outperforming GIT for some of the protein structural classes, highlighting that our individual graphlet-based protein features are capturing at least somewhat complementary information. Because per multiple reviewers' suggestions, we now consider the best of our graphlet features (i.e., OrderedGraphlet-3-4) that is based on a new 6Å (i.e., OrderedGraphlet-3-4 (6Å)) distance cutoff in addition to considering OrderedGraphlet-3-4 that originally used 4Å distance cutoff (i.e., OrderedGraphlet-3-4 (4Å)), and because OrderedGraphlet-3-4 (6Å) significantly outperforms OrderedGraphlet-3-4 (4Å), we now evaluate the complementary nature of GIT versus OrderedGraphlet-3-4 (6Å). So, the statistics have changed. Now, we find that out of 256 different protein structural classes, OrderedGraphlet-3-4 (6Å) outperforms GIT in 84 classes, GIT outperforms OrderedGraphlet-3-4 (6Å) in 132 classes, and the two are tied for the remaining 40 structural classes. That is, OrderedGraphlet-3-4 (6Å) is now at least as good as GIT for almost half of all classes. For details, see Section 3(c).

Second, the *individual* OrderedGraphlet-3-4 (6Å) feature is not the only feature that we propose. We also propose two novel *integrated* features that combine OrderedGraphlet-3-4 (6Å) with existing features. Our new integrated features significantly outperform each of the considered existing features. For details, see Section 3(d).

Comment 12: page 17, line 9-15: Using weighted networks is not better after all?

Our response to comment 12: To remind the reviewer, page 17, line 9-15, of the submitted paper said the following:

“Our proposed DL classifier performs quite well in terms of accuracy (Figs. 4 and 5, and Table 2). Specifically, it is significantly (p -value < 0.05) superior to AAComposition, CSM, and Existingall* (Figure 6). However, the performance of the DL classifier is significantly (p -value < 0.05) lower than OrderedGraphlet3-4, GIT, GIT+OrderedGraphlet-3-4, and Concatenate-all (Figure 6). Yet, the performance of the DL classifier is comparable to one of the two top performing graphlet features, NormOrderedGraphlet-3-4*.”*

We already addressed this comment in our response to comment 1 of reviewer 3.

Comment 13: page 17, line 30-34: What would be a more sophisticated way of weighting the matrix? Can the authors give an example and how these matrices can be used? And why don't the authors do this in their work?

Our response to comment 13: We already addressed this comment as a part of our responses to comments 1 and 4 of reviewer 3. Briefly, there is only one way to weigh a network adjacency matrix. What we refer to “naïve” is how features are extracted from a weighted network. Using a weighted adjacency matrix is a naïve way of doing this. Developing and using e.g., weighted graphlets would be a more sophisticated way to extract features from a weighted network.

Comment 14: page 17, (f): Since SVM can be compared only using 2 out of the 36 sets, should it be kept in the main text? Maybe in the supplement is sufficient?

Our response to comment 14: Done. We assume that the reviewer is referring to the SVM-fold feature, rather than the SVM classifier (which we also mention in the paper). We have now moved most of the discussion and results related to this approach to Supplementary Section S6.

Comment 15: page 18, line 45-56: why don't the authors compare their method using the 6Å cutoff to GIT?

Our response to comment 15: Done. We now do this. That is, we now add results for the best of our graphlet-based features (i.e., OrderedGraphlet-3-4) that is based on the threshold of 6Å to the results that compare all considered methods, including GIT. Because OrderedGraphlet-3-4 with 6Å (i.e., OrderedGraphlet-3-4 (6Å)) significantly outperforms OrderedGraphlet-3-4 with 4Å (i.e., OrderedGraphlet-3-4 (4Å)), we now also update the two of our proposed integrated features that rely on OrderedGraphlet-3-4, i.e., GIT+OrderedGraphlet-3-4 and Concatenate-all, to use the 6Å threshold.

Overall, our conclusions are the same. Our proposed features (either graphlets alone or graphlets integrated with the existing features) still outperform each of the existing considered features. For details, see Sections 3(c) and 3(d).

Comment 16: page 19, line 33-39: I don't see why their method points to a promising future when it does not outperform GIT. Maybe the authors can give an example?

Our response to comment 16: To remind the reviewer, page 19, lines 33-39, in the submitted paper said the following:

“We find that the network-based graphlet features are superior to most of the considered existing features. Additionally, we show that integrating different protein features improves the

PSC accuracy compared to individual features, possibly by capturing complementary protein structural information. Further, our proposed DL framework, which automatically learns appropriate features from simple weighted PSN adjacency matrices, yields comparable accuracy to many of the sophisticated features that we use, which work on unweighted PSNs. This points to a promising future for algorithms that will rely on weighted network-based features of protein 3D structures, such as weighted graphlets, which currently do not exist.”

First, our proposed integrated protein features (that encompass graphlets) do significantly outperform GIT (see Fig. 4 in the main paper). Second, the “promising future” explicitly refers to *future* algorithms that will rely on *weighted* network-based features of protein structures. Our follow-up work on weighted graphlets that has been motivated by the current NETPCLASS paper under consideration already shows the promised improvement. For details, see our point 3 in the Introduction at the beginning of this response letter. Also, for additional information, see our response to comment 4 of reviewer 3.

Comment 17: page 19, line 53-57: Not sure if I would agree here. Do methods exist which are solely applied on protein structure prediction? If so, which? Aren't most also applicable to other problems?

Our response to comment 17: To remind the reviewer, on page 19, lines 53-57, we said the following: *“Second, we believe that studying protein 3D structures as networks is more meaningful than studying the 3D structures directly. This is not just because of the power of networks to reveal interesting data patterns that non network approaches might miss [3, 9, 22, 36], but also because the former allows for a protein structure to be studied with any (current or future) state-of-the-art method for network analysis developed in any field of network science, including the field of protein structural analysis, while the latter allows for using only methods specialized in the field of protein structural analysis.”*

We still stand behind our statement. First, a minor clarification: we say “protein structural *analysis*” and not “protein structural *prediction*”. The latter is a subset of the former. Yes, if a feature uses information that is specific to biochemical properties of a protein structure (and all of AAComposition, di-amino-acid, SVM-fold, CSM, GIT, etc. do), then the only entities that can be studied with such features are protein structures. Importantly, network approaches, including our graphlet approaches that are a state-of-the-art among the network approaches, can be used to study other problems/network data types in the computational biology domain and not just the problem of PSC from protein structures or even just the more general task of protein structural analysis. For example, network approaches can be used for detection of functional modules or interaction prediction in protein-protein interaction networks. Or, they can be used to study a person’s brain network activity while performing a certain task. Also, they can be applied to domains other than computational biology, such as studying human health, spread of people’s behaviors, or targeted advertising in networks corresponding to online social media platforms. Or, they can be used to study robustness of infrastructure networks such as power grids or the Internet. And so on. Clearly, traditional protein structural features that account for e.g., information about amino acids or other biochemical properties of a protein structure are simply not applicable to any problems that rely on entities that are not individual protein structures, such as protein-protein interactions, brain regions, people, power stations, or computing devices. In other words, in any given time period, there will be many more network science methods developed than there will be protein structure-specialized methods developed, because the field of network science is much wider than the field of protein structural analysis. So, if a general network science approach such as graphlets works well in such a specialized task as PSC, any new future superior network science method could advance the field of PSC (and protein structural analysis in general) as well. Bottom line that we are trying to make is that PSN-based analyses of protein structures are powerful.

Comment 18: <https://arxiv.org/abs/1203.2821> : graphlets in weighted networks

Our response to comment 18: Although the arxiv article pointed out by the reviewers deals with weighted networks and uses the term “graphlets”, it does not deal with the concept of graphlets (aka network motifs) as defined in our paper and in the field of network science in general. In fact, as the authors of this paper state: “We term our method ‘graphlet decomposition’, as it is reminiscent of a wavelet decomposition of a graph. An unrelated literature uses the term graphlets to denote network motifs (Przulj et al., 2006)”, where Przulj et al., 2006 are the ones who proposed graphlets as they are defined in our paper. So, this paper is entirely unrelated to our work, i.e., it is not about graphlets in weighted networks. For more details, see our points 3 and 4 from the Introduction of this response letter (on its page 2).

Comment 19: Figure 6: GIT+Ordered does not significantly outperform GIT

Our response to comment 19: Because reviewer 2 suggested several other machine learning-related aspects to further improve our framework and thus fairly compare the different considered features, we have repeated all of our analyses following the recommendations; this includes using a different (better!) version of OrderedGraphlet-3-4, namely the version using PSNs formed with the 6Å distance cutoff, compared to the originally used version of OrderedGraphlet-3-4 at the 4Å cutoff (see our response to comments 1, and 4 of reviewer 2). In summary, we now find that GIT+OrderedGraphlet-3-4 significantly outperforms GIT and every other considered individual protein feature. For details, see Section 3(d).

Comment 20: - The writing could be improved. Some sentences are very complex / hard to read. The abstract is not really smooth

Our response to comment 20: Done. We hope that the entire paper now reads more clearly. But if the reviewer still has any specific follow-up comments on any parts of the text that might remain unclear, we would be happy to address such comments.

Comment 21: *Minor comments.*

Our response to comment 21: We outline each of the minor comments below in italic and then immediately write our response in regular font.

- *Abbreviations: introduce them when the word is used first (e.g. SS page 2 line 37). Introduce all of them (e.g. SVM, page 2 line 38 or HMM page 2 line 37, page 5 line 17 and 20: Abbrev. For PDB)*

Done.

- *don't introduce abbreviations in titles*

Done. We removed abbreviations from section titles.

- *instead of “time complexity” I would use either “running time” or “computational complexity”*

Done. Instead of “time complexity”, we now use “computational complexity (i.e., running time)” the first time we mention computational complexity and from thereafter we use “running time”.

- page 2, line 9: remove dot from title

Done.

- page2, line 14-15: I don't understand the sentence "note existing method."

To remind the reviewer, on page 2, lines 14-15, we said the following: "*note that here we refer to as broad notion of protein structural similarity as captured by any existing method.*"

We modified this sentence as follows to clarify it: "note that here we refer to as broad notion of protein structural similarity as possible, i.e., as captured by any existing method."

- page 2, line 23-31 and lines 46-57: repetition?

Lines 23-31 and lines 46-57 are explaining two different concepts. While lines 23-31 talk about *unsupervised* protein structural *comparison*, lines 46-57 talks about *supervised* protein structural *classification*. So, we left this text as is.

- page 3, line 52 and 53: Twice "propose". Use different word

Done. We now replaced the second "propose" with "suggest".

- page 3, line 56- page 4 line 8: Sentence is hard to understand

To remind the reviewer, on page 3, line 56, we said the following: "*Yet, we hypothesize that the existing PSN definition, which links with unweighted edges those pairs of amino acids whose 3D spatial distance is below some predefined cutoff, can benefit from including as edge weights the actual spatial distances, and by doing so for all pairs of amino acids in the 3D structure rather than only for those pairs that are below the given distance cutoff.*"

Thank you for catching this extremely complex sentence. We have now simplified it as follows: "Existing PSN definition links with unweighted edges those pairs of amino acids whose 3D spatial distance is below some predefined cutoff. Current graphlets can deal with such edge-unweighted networks. We hypothesize that PSC can benefit from including the actual spatial distances as edge weights."

- page 4, line 53-56: does a table with running times exist? If so, refer to it. If not, create one

Yes. We do have figures with running times in the Results section. However, at this point in the paper, we are in the Introduction section and hence we do not show any results yet.

- page 5, line 15: why bold?

Because the first and second points are far apart in the paper, a reader might want to revisit the first point when she/he reads the second point. In that case, the reader might need to back to the first point. Hence, we made the first point bold to facilitate this.

- page 8, line 6: typo: "If only if". There is something missing

This is a typo, it should have been “only if”, we have corrected this now.

- page 8, line 35-40: Restructure the sentence. It is not nicely written

To remind the reviewer, on page 8, lines 35-40, we said the following: “*Because AAComposition considers the entire protein sequence, it is a global feature. Because AAComposition does not use any heavy atom of an amino acid, it is neither a backbone- nor a side-chain-based feature. Because AAComposition only measures the relative frequency of the different amino acid types without looking at the sequence position of a given amino acid, it is not sequence-dependent.*”

Former lines 35-40 encompass three sentences. We believe the sentences to be grammatically correct and understandable. In each sentence, the “because” part offers a fact, and that fact justifies whether the feature is global (sentence 1), backbone- or side-chain-based (sentence 2), and sequence-dependent (sentence 3). Nonetheless, to address the reviewer’s comment, we modified these sentences as follows to clarify them: “AAComposition considers the entire protein sequence, and hence, it is a global feature. AAComposition does not use any heavy atom of an amino acid, and hence, it is neither a backbone- nor a side-chain-based feature. AAComposition only measures the relative frequency of the different amino acid types without looking at the sequence position of a given amino acid, and hence, it is not sequence-dependent.” In other words, from the “because x, y” form, we modified into “x, and hence, y” form. We hope that this makes the sentences easier to read and understand.

- page 8, line 44-49: Restructure the sentence. It is not nicely written

To remind the reviewer, on page 8, lines 44-49, we said the following: “*Because SVMfold relies on HMM profile, which extracts features from subsequences of a protein, SVMfold is a local feature. Because SVMfold does not explicitly use any heavy atom of an amino acid, it is neither a backbone- nor a side-chain-based feature. Because SVMfold relies on PSSM profile, which extracts features from a sequence alignment of proteins, SVMfold is sequence-dependent.*”

We have modified the sentences in the similar manner as we do for the previous comment above.

- page 9, line 10-15: Restructure the sentence. It is not nicely written

To remind the reviewer, on page 9, lines 10-15, we said the following: “*Because GIT measures local patterns formed by the α -carbon trace of a protein, it is a local feature. Because GIT is based on only the α -carbons of a protein and because α -carbons are part of the backbone of a protein, GIT is a backbone-based feature. Because GIT is based on the α -carbon trace of a protein and because a change in sequence positions of amino acids might affect the α -carbon trace of a protein, GIT is sequence-dependent*”

We have modified the sentences in the similar manner as we do for the previous two comments above.

- page 9, line 18-25: Restructure the sentence. It is not nicely written

To remind the reviewer, on page 9, lines 18-25, we said the following: “*For a given distance cutoff, two amino acids are considered to be in contact if any of their heavy atoms are within the distance cutoff; then, CSM counts the number of amino acid pairs that are in contact at that cutoff. Finally, CSM uses all 151 counts (for the 151 cutoffs) as the 151-dimensional protein feature vector [32]. Because CSM counts the total number of contacts present in the whole protein, it is a global*

feature. Because CSM counts the number of contacts and because a change in sequence positions of amino acids without altering 3D positions of the amino acids will not alter the number of contacts, CSM is not sequence-dependent.”

We have modified the sentences in the similar manner as we do for the previous three comments above.

- page 9, line 31-34: Restructure the sentence. It is not nicely written

To remind the reviewer, on page 9, lines 31-34, we said the following: “*Because Existing-all uses global network measures to quantify the structure of a protein (i.e., PSN), it is a global feature. Because Existing-all extracts features from a PSN and because a change in sequence positions of amino acids without altering 3D positions of the amino acids will not alter the PSN structure, Existing-all is not sequence-dependent.”*

We have modified the sentences in the similar manner as we do for the previous four comments above.

- page 9, line 40-44: Restructure the sentence. It is not nicely written

To remind the reviewer, on page 9, lines 40-44, we said the following: “*Because graphlets are small network patterns of a PSN and because Graphlet-3-4 and Graphlet-3-5 are based on graphlets, both Graphlet-3-4 and Graphlet-3-5 are local features. Because both Graphlet-3-4 and Graphlet-3-5 extract features from a PSN and because a change in sequence positions of amino acids without altering 3D positions of the amino acids will not alter the PSN structure, both Graphlet-3-4 and Graphlet-3-5 are not sequence-dependent.”*

We have modified the sentences in the similar manner as we do for the previous five comments above.

- page 10, line 8: Underline “NormOrderedGraphlet-3” and “NormOrderedGraphlet-3-4” too for consistency

These are actual names of the features and not categories of features. So, NormOrderedGraphlet-3 and NormOrderedGraphlet-3-4 as presented in the paper should not be underlined; we only underlined categories of features.

- page 10, line 38-40: Restructure the sentence. It is not nicely written

To remind the reviewer, on page 10, lines 38-40, we said the following: “*Because ordered graphlets are essentially graphlets, all of the ordered graphlet-based features are local. Because ordered graphlets use node order as per sequence positions of amino acids in a protein and because a change in sequence positions of amino acids might change ordered graphlet counts, all of the ordered graphlet-based features are sequence-dependent.”*

We have modified the sentences in the similar manner as we do for the second last comment above.

- page 10, line 43: dot after “)”

Done.

- page 10, line 46: Remove abbrev. Or “linear regression”

Removed.

- page 11, line 10: *No intro of abbrev. In title*

Removed.

- page 11, line 24: *Use SVM instead of “support vector machine”*

Done.

- page 11, line 47: *Use DL instead of “deep learning”*

Done.

- page 18, line 23: *“joins” is misleading. I would use “connects”*

Done. Replaced “joins” with “connects”

- page 20, line 6-9: *Very vague sentence and not really proven. I would recommend to remove it*

To remind the reviewer, on page 10, lines 38-40, we said the following: *“And network approaches, given a large number of other network approaches acting as their competitors, need to go through a thorough evaluation and continuous improvements. This should continue to result in more and more powerful network approaches.”*

We agree with the reviewer and have consequently removed this sentence.

- table1: *why is there no vertical line in the last row?*

Added.

- table 3: *Make figure to this table?*

Table 3 showed results related to the SVMfold method. Because we were able to apply SVMfold to only 2 out of the considered 36 PSN sets, as suggested by the reviewer, we have moved that section of the paper, along with table 3, to the Supplementary Section S6 (Supplementary Table S13). Consequently, we believe that there is no need for yet another supplementary figure, as the table already provides the full information.